# Active Curriculum Refinement for Reinforcement Learning

**Zhenya Liu** [1]    **Yuxin Chen** [1]

## Abstract

In many RL domains, environments are linked by prerequisite relations—e.g., difficulty-increasing edits or parameter increments—which induce a directed acyclic curriculum graph (DAG). In practice, this structure is often exploited only implicitly, yet it can yield clear gains in training. We introduce PATH, a curriculum learning framework that performs active learning on the curriculum graph. PATH first expands coverage by sampling diverse curriculum paths, then reallocates training toward regions that remain unmastered. Experiments show that PATH leverages the graph structure to achieve strong robustness and generalization across diverse environments.

## 1. Introduction

Deep reinforcement learning (RL) is a central paradigm in modern AI, underpinning successes in complex decision-making problems (Silver et al., 2016) and post-training for instruction-following LLMs (Ouyang et al., 2022). As RL systems move beyond single benchmark instances, a core challenge is generalization: policies trained on a finite set of environments often overfit and fail to transfer robustly across a broader environment distribution (Cobbe et al., 2019). Curriculum learning (Bengio et al., 2009; Wang et al., 2022) offers a practical approach for RL generalization by controlling which environments are visited and in what order (Narvekar et al., 2020; Parker-Holder et al., 2022; Soviany et al., 2022), and recent work increasingly focuses on autonomously generated curricula, including unsupervised environment design (UED) (Dennis et al., 2020). While fully automatic curriculum generation is often emphasized, many effective methods do not build curricula from scratch; instead, they implicitly exploit pre-existing structure in the environment distribution—e.g., an inherent easier-to-harder ordering that supports curriculum progression (Florensa et al., 2017; Parker-Holder et al., 2022).

Crucially, in many RL settings such curriculum orderings are straightforward to specify, and can be made explicit in three practical ways. (i) *Parameterized increments:* environments expose interpretable parameters, so successor environments can be generated by small increases along a chosen parameter (Wang et al., 2019; Tidd et al., 2021). (ii) *Complexity-increasing edits:* successor environments are obtained by local edits that make the instance more complex, e.g., adding obstacles (Li et al., 2020; Parker-Holder et al., 2022). (iii) *Feature-induced structure:* when parameters or edits are unavailable, instances can be represented by descriptors or embeddings, and edges follow monotone progress along one or multiple feature axes (Lee et al., 2025). In all three cases, the environment space provides usable curriculum orderings, naturally inducing a directed acyclic graph (DAG) over instances; training along a single ordering often yields an effective curriculum progression.

This raises a natural question: given an explicit curriculum DAG, can we allocate training efficiently so that traversing relatively few paths suffices to master the entire reachable environment space? We formalize this as a new curriculum learning framework—*active learning on the curriculum graph*. We introduce PATH, a two-stage automated and adaptive algorithm: PATH:RANDOM samples curriculum paths randomly to drive broad exploration and expand coverage, and PATH:ACTIVE uses regret-based signals to identify high learning-potential regions and reallocates budget by sampling more paths through their successor subgraphs. Overall, PATH treats curriculum learning as active path acquisition on a curriculum DAG, enabling efficient mastery of the reachable environment space. In summary, our contributions are:

- We formalize curriculum learning over structured environment families as active learning on a curriculum DAG, where training corresponds to acquiring informative paths rather than isolated instances.

- We introduce PATH, a lightweight two-stage algorithm that combines random path exploration for coverage with regret-driven successor expansion to efficiently allocate training budget.

---

[1]The University of Chicago, Chicago IL, USA. Correspondence to: Zhenya Liu <zhenya@uchicago.edu>, Yuxin Chen <chenyuxin@uchicago.edu>.

*Proceedings of the 43rd International Conference on Machine Learning*, Seoul, South Korea. PMLR 306, 2026. Copyright 2026 by the author(s).

- We demonstrate on both discrete (MiniGrid) and continuous-control (BipedalWalker) benchmarks that exploiting curriculum graph structure yields substantially improved robustness and generalization over prior curriculum and regret-based methods.

# 2. Preliminaries

## 2.1. Reinforcement Learning

**Environment.** An environment is a discounted Markov decision process (MDP) $\mathcal{M} = (\mathcal{S}, \mathcal{A}, P, r, \rho_0, \gamma)$, where $\mathcal{S}$ is the state space, $\mathcal{A}$ is the action space, $P(\cdot \mid s, a)$ is the transition kernel, $r(s, a, s')$ is the reward function, $\rho_0$ is the initial-state distribution, and $\gamma \in (0, 1)$ is the discount factor. This definition corresponds to a fully specified environment instance with fixed dynamics and rewards.

**Partially Observable MDP (POMDP).** To model partial observability, we consider a partially observable MDP (POMDP), defined as $\mathcal{M} = (\mathcal{S}, \mathcal{A}, \mathcal{O}, P, r, \mathcal{I}, \rho_0, \gamma)$, where $\mathcal{O}$ is the observation space and $\mathcal{I}(\cdot \mid s)$ is the observation function mapping latent states to observations. The underlying environment dynamics $(P, r)$ remain fixed, but the agent only observes $o_t \sim \mathcal{I}(\cdot \mid s_t)$.

**Underspecified POMDP.** Following Dennis et al. (2020), we model learning across environments using an *underspecified* POMDP (UPOMDP), $\mathcal{M} = (\mathcal{S}, \mathcal{A}, \mathcal{O}, \Theta, P, r, \mathcal{I}, \rho_0, \gamma)$, where $\Theta$ is a set of environment instances. The transition kernel and reward depend on $\theta$, written as $P : \mathcal{S} \times \mathcal{A} \times \Theta \to \Delta(\mathcal{S})$ and $r : \mathcal{S} \times \mathcal{A} \times \mathcal{S} \times \Theta \to \mathbb{R}$. Fixing $\theta$ yields a concrete POMDP $\mathcal{M}_\theta = (\mathcal{S}, \mathcal{A}, \mathcal{O}, P_\theta, r_\theta, \mathcal{I}, \rho_0, \gamma)$. We identify an instance $\theta \in \Theta$ with its induced POMDP $\mathcal{M}_\theta$ (i.e., we use $\theta$ and $\mathcal{M}_\theta$ interchangeably in the remainder of the paper).

**Learning Objective.** Given an underspecified POMDP characterized by a set of environments $\Theta$, our learning objective is applying reinforcement learning (Sutton & Barto, 2018) to find a policy that performs optimally across all environments. Let $\Pi$ denote the policy class, and let $V(\pi, \theta) := \mathbb{E}_{\rho_0, \pi, P^\theta}[\sum_{n=0}^\infty \gamma^n r_n]$ denote the expected discounted return of policy $\pi \in \Pi$ in environment $\mathcal{M}_\theta$. The regret of policy $\pi$ on environment $\theta$ is defined as

$$\mathrm{Regret}(\pi, \theta) := -V(\pi, \theta) + \max_{\pi' \in \Pi} V(\pi', \theta),$$

which measures the suboptimality of $\pi$ relative to the optimal policy for $\theta$.

Our goal is to learn a policy $\pi^\star \in \Pi$ that performs well across the entire environment set $\Theta$ induced by the UPOMDP. Ideally, this corresponds to the objective $\pi^\star \in \arg\min_{\pi \in \Pi} \min_{\theta \in \Theta} \mathrm{Regret}(\pi, \theta)$ which captures the goal of achieving low regret over all environment instances. Throughout the paper, we refer to eliminating regret on an environment instance as *mastering* that environment.

## 2.2. Curriculum of Environments as a Graph

We make the natural ordering structure exploited by many curriculum methods—discussed in Section 1—explicit by representing the curriculum as a directed acyclic graph (DAG) over the environment space $\Theta$ (Svetlik et al., 2017; Narvekar et al., 2020).

**Definition 2.1** (Curriculum DAG). A curriculum is a directed acyclic graph $\mathcal{G} = (\Theta, \mathcal{E})$, where each node corresponds to an environment instance $\theta \in \Theta$, and each directed edge $(\theta, \theta') \in \mathcal{E}$ indicates that training on $\theta$ is intended to facilitate learning on $\theta'$.

A directed path $P = (\theta_1, \ldots, \theta_L)$ in $\mathcal{G}$ defines a curriculum sequence, where environments are trained in the order specified by the path. A path is maximal if it starts from a source node (in-degree 0) and ends at a sink node (out-degree 0). Let $\mathcal{P}(\mathcal{G})$ denote the set of all maximal paths in the curriculum DAG.

**Curriculum as prerequisite structure.** Along any maximal path $P = (\theta_1, \ldots, \theta_L) \in \mathcal{P}$, we assume that respecting the path order enables efficient progression: there exists a constant $m > 0$ such that for each $\theta_i$ on the path, once $\theta_{i-1}$ is mastered, allocating at most $m$ effective training interactions on $\theta_i$ suffices to learn $\theta_i$ This captures the core advantage of structured curricula over unstructured training: prerequisite dependencies reduce the effective cost of mastering harder instances compared to treating environments as independent.

# 3. Proposed Methods

We propose a new algorithm PATH that exploits curriculum structure defined over a directed acyclic graph (DAG) on the set $\Theta$. The method consists of two components: a randomized path exploration procedure (PATH:RANDOM) and a regret-driven active learning procedure (PATH:ACTIVE).

## 3.1. PATH:Random

PATH:RANDOM performs stochastic curriculum exploration by maintaining $N$ path pointers on a curriculum DAG $\mathcal{G}$ over environments $\Theta$. Each pointer identifies a current environment $\theta \in \Theta$. At each iteration, we uniformly sample a batch of pointers and update the student policy $\pi$ using rollouts collected in their current environments. Equivalently, this is uniform sampling from a dynamic buffer $\mathcal{B}$ of size $N$, where each entry is the environment currently pointed to by a path.

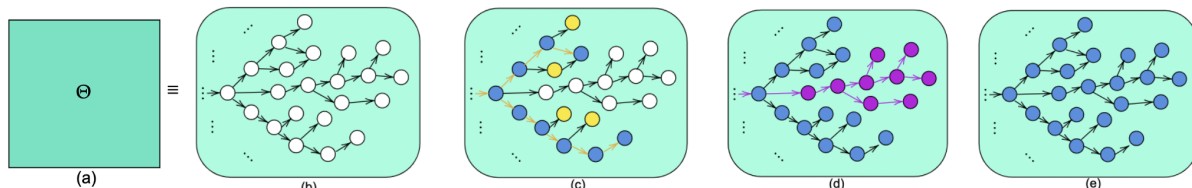

*Figure 1.* Overview of PATH. (a) A environment distribution $\Theta$. (b) The same space viewed as an explicit curriculum DAG $\mathcal{G}$, where edges encode prerequisite relations between environments. (c) PATH:RANDOM explores $\mathcal{G}$ by sampling random paths (yellow trajectories); blue nodes are explicitly visited and learned, while yellow nodes denote additional environments learned via generalization from nearby visited nodes. (d) PATH:ACTIVE identifies a node with high learning potential, then it allocates more sampling by launching multiple random paths (purple) from that node. (e) This adaptive allocation rapidly increases coverage and efficient learns all nodes in $\mathcal{G}$.

Pointers advance via on-the-fly path generation, avoiding explicit enumeration of $\mathcal{G}$ whenever successors can be constructed from the current pointer via local edits or parameter increments. Each path pointer starts from a uniformly sampled source node. When a path pointer advances, it samples one outgoing edge uniformly and moves to the successor. If it reaches a sink, it is restarted from a new uniformly sampled source. We refer to this path sampling method as *random path sampling*. Advancement is controlled by two stopping rules. (1) *Early-stop reward* $\varepsilon_{\mathrm{es}}$: In each policy update, for every sampled environment $\theta$ we collect $K_\theta$ episodes with episodic returns $\{R_{\theta,1}, \ldots, R_{\theta,K_\theta}\}$, and compute the mean return $\bar{R}_\theta = \frac{1}{K_\theta} \sum_{k=1}^{K_\theta} R_{\theta,k}$. If $\bar{R}_\theta \geq \varepsilon_{\mathrm{es}}$, we treat $\theta$ as solved and advance immediately. (2) *Sampling patience* $\varepsilon_{\mathrm{p}}$: if $\theta$ has been sampled $\varepsilon_{\mathrm{p}}$ times without meeting $\varepsilon_{\mathrm{es}}$, we terminate and reinitialize the path. In practice, $\varepsilon_{\mathrm{p}}$ is set to approximate the sample budget $m$ required to efficiently learn any $\theta$ along the curriculum path. We defer the sensitivity analysis over $\varepsilon_{\mathrm{es}}, \varepsilon_{\mathrm{p}}$ to Appendix C.4.2.

### 3.2. PATH:Active

PATH:ACTIVE replaces uniform sampling with regret-weighted replay over the current path pointers. We maintain $N$ curriculum trajectories in parallel and collect their current pointer environments into a dynamic buffer $\mathcal{B} \subset \Theta$. Each $\theta \in \mathcal{B}$ is assigned a nonnegative weight $w(\theta)$ (initialized uniformly). At each iteration, we sample a training batch from $\mathcal{B}$ proportionally to $w(\theta)$. For each sampled $\theta$, we use rollouts in $\theta$ to collect TD-error $\{\delta_t^\theta\}$ at each time $t$. We update $w(\theta)$ using the regret proxy–*postive value loss* used by PLR (Jiang et al., 2021b):

$$S(\theta) := \frac{1}{T} \sum_{t=0}^{T} \max\left( \sum_{k=t}^{T} (\gamma\lambda)^{k-t} \delta_k^\theta, 0 \right),$$

where $\lambda$ and $\gamma$ are the *Generalized Advantage Estimation* (Schulman et al., 2016) hyperparameters. Higher $S(\theta)$ indicates that the policy's most recent interactions with $\theta$ yielded unexpectedly high returns, suggesting strong learning potential.

We use the same early-stop rule as PATH:RANDOM: if the policy's recent reward on $\theta$ reaches $\varepsilon_{\mathrm{es}}$, we advance by sampling an outgoing edge from $\theta$ in $\mathcal{G}$. When this happens, we also expand exploring from $\theta$ by inserting a batch of its successor pointers into the buffer $\mathcal{B}$, so subsequent sampling dispatches multiple paths through the successor region. All newly advanced pointers are inserted into $\mathcal{B}$; to maintain $|\mathcal{B}| = N$, we evict low-weight pointers so the buffer stays concentrated on high regret regions while still tracking curriculum progress. At each iteration, with the replay rate $p$, we choose between (i) sampling a batch of pointers from $\mathcal{B}$, or (ii) inserting an initialized random batch of source nodes into $\mathcal{B}$. Injecting new start nodes serves two purposes: (i) sampling them leads to revisit explored paths—if a node already meets the $\varepsilon_{\mathrm{es}}$, we advance immediately without wasting training budget; otherwise, additional updates help the policy relearn it and mitigate catastrophic forgetting (Rolnick et al., 2019); and (ii) new added nodes cause evicting low-regret pointers, which increases the turnover of paths tracked in $\mathcal{B}$, speeding up the discovery of high-regret regions.

### 3.3. PATH

PATH is a two-stage algorithm. We start with the random phase, which efficiently advances along diverse curriculum paths. Let $n$ denote the number of maximal curriculum paths that have terminated. We check the switch condition $n \geq T_{\mathrm{switch}}$ to ensure a more consistent switch point across runs; once satisfied, we switch to PATH:ACTIVE.

At the transition, we continue training with PATH:ACTIVE on the same buffer. Intuitively, $T_{\mathrm{switch}}$ is chosen so that the random stage has already achieved broad exploration; beyond this point, path terminations and revisits become frequent, so reallocating sampling toward high-learning-potential regions is more effective. Figure 1 provides an overview of PATH and its two-stage training procedure. We provide one simplified version of PATH in Algorithm 1. Full algorithmic details for all three algorithms are provided in Appendix A.

**Algorithm 1** PATH

1: **Input:** curriculum DAG $\mathcal{G}$, pointer buffer size $N$, replay rate $p$, weights $w(\cdot)$, early-stop threshold $\varepsilon_{\text{es}}$, sampling patience $\varepsilon_{\text{p}}$, transition time $T_{\text{switch}}$
2: **Initialize:** policy $\pi(\phi)$; fill buffer $\mathcal{B}$ with source nodes of $\mathcal{G}$; initialize weights $w$ on $\mathcal{B}$
3: **while** not converged **do**
4:    **if** $n \leq T_{\text{switch}}$ **then**
5:       Sample node $\theta \sim \text{Unif}(\mathcal{B})$ and train $\pi$ on $\theta$
6:       Change $\theta$ to one of its successor in $\mathcal{G}$ if $\varepsilon_{\text{es}}$ meets, drop it if $\varepsilon_{\text{p}}$ meets
7:       If $\theta$ drops, add one source node from $\mathcal{G}$ to $\mathcal{B}$
8:       $n \leftarrow n + 1$
9:    **else**
10:      Sample decision $d \sim \text{Bernoulli}(p)$
11:      **if** $d = 1$ **then**
12:        Sample node $\theta \sim \mathcal{B}$ with probability $\propto w(\theta)$
13:        Train $\pi$ on sampled $\theta$ and update $w(\theta) \leftarrow S(\theta)$
14:        Add one $\theta$ successor in $\mathcal{G}$ to $\mathcal{B}$ if $\varepsilon_{es}$ meets
15:      **else**
16:        Sample a source node of $\mathcal{G}$, add to $\mathcal{B}$
17:      Evict low $w$ nodes from $\mathcal{B}$ to keep $|\mathcal{B}| = N$

### 3.4. From Random to Active Path Allocation

We now motivate the necessity of the two phases of PATH. We formulate our learning goal as mastering an *unknown* reachable subgraph $\bar{\mathcal{G}} \subseteq \mathcal{G}$ induced by the policy's current capability. $\varepsilon_{\text{es}}$ determines how far the policy can progress along any curriculum path: once a node fails to reach $\varepsilon_{\text{es}}$, the successor region beyond that node becomes effectively unreachable at the current stage. Thus, the leaves of $\bar{\mathcal{G}}$ mark the policy's current frontier in the environment space $\Theta$. We measure progress by *generalization coverage* over $\bar{\mathcal{G}}$.

**Generalization coverage.** Let $\bar{\mathcal{G}} = (V, E)$. For a maximal curriculum path $P$, let $S(P) \subseteq V$ be its visited nodes. We use a graph-local neighborhood $B_t(u) = \{v \in V : d(u,v) \leq t\}$ (where $d(u,v)$ is number of edges of the shortest path from $u$ to $v$ in undirected version of $\bar{\mathcal{G}}$) to represent the maximal region that *could* be influenced by training on $u$. The *realized* generalization is modeled as an unknown subset $G(u) \subseteq B_t(u)$, i.e., the set of nodes that become mastered due to generalization when training on $u$ under the current policy. We do not assume $G(u)$ fills the entire $t$-ball, as generalization is typically partial and depends on specific nodes. A path induces coverage $C(P) = \bigcup_{u \in S(P)} G(u)$, and for $n$ sampled paths $\mathcal{P}_n = (P_1, \ldots, P_n)$ the global coverage is $f(\mathcal{P}_n) = \left| \bigcup_{i \leq n} C(P_i) \right|$. Our goal is to expand $f(\mathcal{P}_n)$ over $\bar{\mathcal{G}}$ using as few sampled paths as possible.

**PATH:RANDOM: growing low-overlap neighborhoods.** In the random phase, we sample paths by choosing a child uniformly at each step, inducing a distribution $\pi$ over maximal paths. Draw $P_1, \ldots, P_n \overset{iid}{\sim} \pi$ and define the marginal gain $\Delta_k := f(\mathcal{P}_k) - f(\mathcal{P}_{k-1})$, where $\Delta_k = \left| C(P_k) \setminus \bigcup_{i<k} C(P_i) \right|$. Since $G(u) \subseteq B_t(u)$, each cover set satisfies $C(P) \subseteq \bigcup_{u \in S(P)} B_t(u)$. Therefore, in an early regime where the $t$-balls encountered by different random paths have limited overlap, the cover sets $\{C(P_i)\}$ also overlap little and $\Delta_k$ remains large, yielding stable growth of $f(\mathcal{P}_n)$. This regime is promoted by the curriculum graph with rich branching and weak bottlenecks, where independent random walks diverge quickly and neighborhood overlap accumulates slowly as more paths are sampled.

As $n$ continues growing, $\bigcup_{i \leq n} \bigcup_{u \in S(P_i)} B_t(u)$ begins to saturate: additional random paths increasingly revisit previously covered neighborhoods, so newly visited nodes $u$ typically have $B_t(u)$ largely contained in the existing union. When $P_k$ visits new nodes, they often lie inside the saturated neighborhood and contribute little new mass to $f(\mathcal{P}_n)$. More importantly, because realized generalization is a subset $G(u) \subseteq B_t(u)$, the remaining unmastered points $B_t(u) \setminus G(u)$ can form a small residual inside visited $t$-balls. Randomly sampling additional paths then becomes an inefficient way to repeatedly hit this residual.

**PATH:ACTIVE: allocating budget to the residual area.** The active phase targets precisely this residual regime. Using regret signals, we identify under-mastered nodes with high learning potential and allocate additional budget by dispatching multiple paths from these anchors, rather than continuing uniform exploration over the whole graph. This exploits prerequisite structure of the DAG curriculum: we assume a high regret node implies that its successor region contains further under-mastered nodes. Concentrated sampling increases the chance of repeatedly training on the residual portion of $G(\cdot)$ that random exploration hits only rarely. However, the same concentration explains weaker *early* coverage for purely active sampling: dispatching many paths from a few starters makes visited $t$-balls highly overlapping, so the neighborhood grows slowly at the beginning and yields less broad coverage than diversified random paths in the early stage. Moreover, in the early regime where regret is high for most entries in $\mathcal{B}$, the fixed replay rate in PATH:ACTIVE—inserting newly random source nodes and evicting low-regret pointers—cannot reliably isolate truly low regret entries. As a result, it may evict paths that are still high-regret but under-trained, prematurely truncating progression and slowing early learning.

Overall, PATH leverages two complementary mechanisms: (i) PATH:RANDOM rapidly expands the union of (mostly non-overlapping) $t$-balls early, yielding large coverage gains; and (ii) PATH:ACTIVE later reallocates training budget to the residual unmastered mass, yielding steadier improvement once random exploration becomes inef-

ficient. We illustrate the predicted behaviors of the three algorithms in Figure 2, and give the empirical evidence to support our analysis in section 4.3.

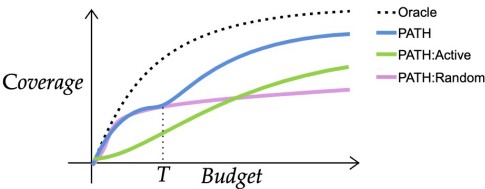

*Figure 2.* Conceptual comparison of training budget versus curriculum graph coverage. Oracle denotes the theoretical optimum. PATH switches at update $T$.

# 4. Experiments

We evaluate PATH against classical curriculum baselines from unsupervised environment design, focusing on methods that construct curricula solely through sampling of the environment set $\Theta$ [1]. In all experiments, the student agent is trained using Proximal Policy Optimization (PPO) ((Schulman et al., 2017)). To isolate the effect of curriculum design, we report performance primarily as a function of the number of student policy-gradient updates (with each update computed from a fixed number of environment steps), which is comparable across methods.

Our baselines include DR (Tobin et al., 2017), PLR (Jiang et al., 2021b;a), and ACCEL (Parker-Holder et al., 2022). Among them, ACCEL serves as the strongest replay-based baseline under this setting. These baselines span increasingly structured sampling: DR uses uniform randomization, PLR performs score-based replay, and ACCEL/PATH incorporate active selection within curriculum structure. ACCEL mutates an environment immediately after sampling and does not explicitly enforce progression along a single curriculum path. In contrast, PATH formulates training as active learning over a curriculum graph, advancing along a path only after the preceding environment meets the mastery criterion. For hyperparameters and compute statistics, see Section B.4.

We evaluate on two highly representative RL testbeds that admit general purpose curriculum constructions. The first one, MiniGrid (Chevalier-Boisvert et al., 2018), instantiates sparse-reward, discrete control, where curricula can be defined as a topologically contiguous sequence of environments obtained by incremental edits (e.g., progressively adding obstacles), yielding an explicit notion of curriculum progression. The second evaluated task, Bipedal-Walker (Brockman et al., 2016), instantiates dense-reward, continuous control, where curricula are most naturally

specified in an environment-parameter space by monotonically increasing difficulty (e.g., harder terrain configurations). These benchmarks cover broad classes of practical RL problems, and the curriculum constructions we use—topologically contiguous edits and monotone difficulty schedules—are generic templates that apply to real world applications. For each environment, we evaluate both robustness and generalization, using metrics such as IQM (interquartile mean) (Agarwal et al., 2021), mean return, and solved rate. We additionally evaluate PATH on Leaper from Procgen (Cobbe et al., 2020), an implicit-parameterization benchmark where difficulty is inferred from observable environmental axes rather than explicit parameters (Appendix C.3).

## 4.1. MiniGrid

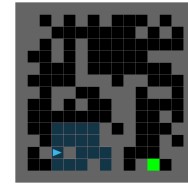
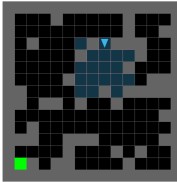
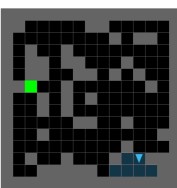

*Figure 3.* Examples of MiniGrid layouts from generalization testset that PATH solves.

We first consider a partially observable navigation task (Dennis et al., 2020) based on MiniGrid. The environment set $\Theta$ consists of all $15 \times 15$ maze layouts obtained by placing between 0 and 60 blocks in an initially empty room. The agent starts in one location and aim to find the goal. We define a curriculum DAG over $\Theta$, in which each node corresponds to a particular room layout and each directed edge denotes an edit that adds blocks. For a fair comparison, we mirror the implicit curriculum induced by ACCEL's editing procedure (Parker-Holder et al., 2022): each curriculum path begins from a randomly generated empty room, and each transition applies a single edit that adds a random number of blocks uniformly in [1,5]. A path terminates once the layout reaches the maximum complexity of 60 blocks.

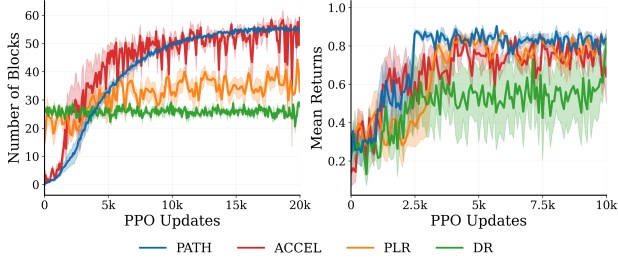

*Figure 4.* Left: emergent complexity of sampled MiniGrid layouts during training. Right: average episodic return on layouts sampled during training. Curves show mean ± standard error across 5 training seeds.

---

[1]Our code is available at https://github.com/Liu-Zhenya/PATH.

Full environment and curriculum specifications are provided in Appendix B.1. Agents are trained for 20k PPO updates using an LSTM-based policy operating on partially observable inputs. We find that, up to 20k updates, PATH remains in PATH:RANDOM and continues to make steady progress, with no evidence of stagnation. In Figure 4, we report the sampling-complexity metric alongside mean training returns. PATH samples complex layouts stably and reaches a higher training return in fewer updates than all other methods.

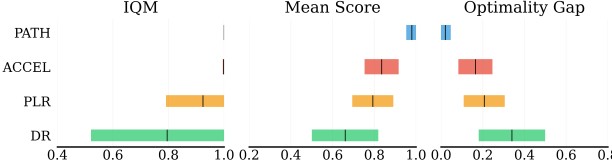

*Figure 5.* Aggregate robustness performance (solved rates) in MiniGrid. The vertical line indicates the mean, and the colored bar shows ±1 standard error.

We assess robustness by evaluating the solved rate by each method on ten hand-crafted challenging environments. The complete set of robustness test layouts and per-environment results are reported in Figure 12 and Table 4 in appendix C.1.1. PATH solves the majority of these test environments As summarized in the aggregate robustness statistics (Figure 5), while ACCEL also achieves near-perfect IQM, PATH attains a mean score of up to 98%, whereas ACCEL and PLR are around 80%.

*Table 1.* Generalization test solved rates at 20k updates (mean ± standard error) for 5 runs of each method on MiniGrid. Bold values are within one standard error of the best mean.

| DR | PLR | ACCEL | PATH |
|---|---|---|---|
| $0.950 \pm 0.0$ | $0.977 \pm 0.0$ | $0.976 \pm 0.0$ | $\mathbf{0.994} \pm 0.0$ |

We further evaluate generalization by sampling 1000 random layouts from $\Theta$. Table 1 shows that all methods exceed 95% solved rate, with PATH achieving a near-perfect 99.4% solved rate, indicating strong generalization. Crucially, PATH performs well on both robustness and generalization, which we attribute to its broad coverage of the environment space. Figure 3 shows representative layouts that are solved by PATH after 20k updates; additional details of the generalization evaluation are provided in Appendix C.1.2. Notably, the random exploration stage already reaches near-perfect performance across the space, which we attribute to broad coverage from diverse random paths coupled with curriculum progression.

## 4.2. BipedalWalker

We next evaluate on the modified version of Bipedal-Walker (Wang et al., 2019). Following the definition from Parker-Holder et al. (2022), each environment is parame-

terized by a 8-dimensional vector $\theta$ that specifies terrain attributes (e.g., gaps, stumps, stairs, and surface roughness), so the environment family induces a continuous, multi-dimensional space $\Theta$. We discretize $\Theta$ from a designated easy start configuration under the same editor used by ACCEL. Concretely, each curriculum path begins at a simple terrain and advances complexity such as adding stumps, stairs by incrementing one or more coordinates of $\theta$. Full details of the parameterization, discretization, and curriculum construction of BipedalWalker are provided in Appendix C.2.

Similar to MiniGrid, each algorithm is evaluated at 20k PPO updates using robustness and generalization metrics.

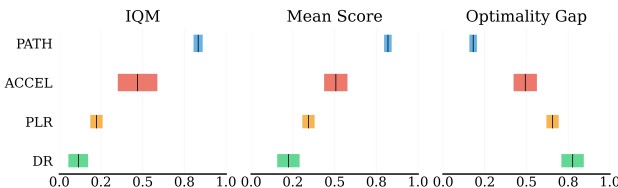

*Figure 7.* Aggregate normalized robustness performance (returns) in BipedalWalker. Returns are min–max normalized from $[-100, 300]$ to $[0, 1]$. The vertical line indicates the mean, and the colored bar shows ±1 standard error.

To evaluate robustness, we adopt the held-out test environment classes from Parker-Holder et al. (2022): HARD-CORE, ROUGHNESS, STAIRS, PITGAP, and STUMP. We evaluate 128 fixed environments per class for each method, and examples of each class are given at Figure 14 in Appendix C.2.1. Figure 6 reports per-class performance over training, and Figure 7 summarizes aggregate normalized-return statistics at the 20k checkpoint across classes. Overall, PATH consistently outperforms the baselines on every robustness class, reaching approximately 90% in both IQM and mean normalized return at 20k updates, whereas ACCEL remains below 60%. Full class-wise results at 20k are reported in Table 5 in Appendix C.2.1.

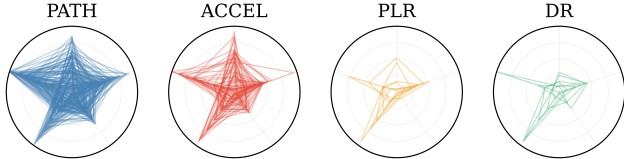

*Figure 8.* Radar plots of terrain parameters for all solved Bipedal-Walker terrains from a single run of each algorithm. Axes are ordered clockwise starting at 12 o'clock: `Roughness` (max 10), `Gap` (max 10), `Stump` (max 5), `Stairs` (max 5), and `Steps` (max 9). Each polyline corresponds to one solved terrain.

To test generalization, we uniformly sample 1000 configurations from $\Theta$ to form a fixed test set (Appendix C.2.2), and report each method's solved rate in Table 2. Although most configurations in $\Theta$ are unsolvable for the trained pol-

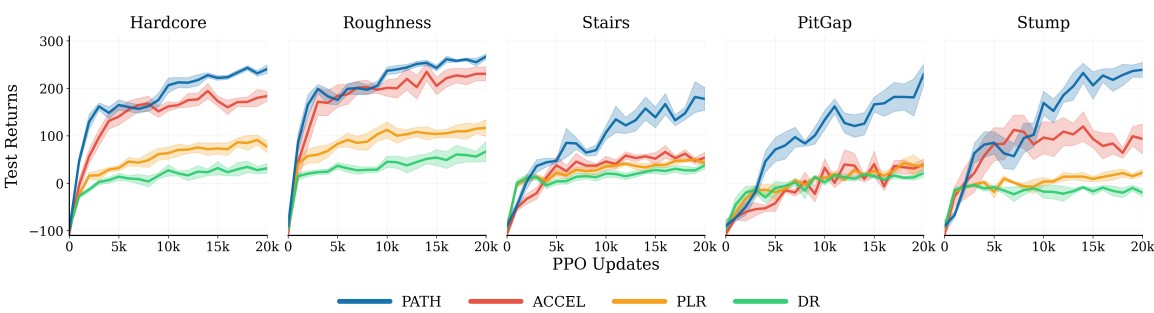

*Figure 6.* Performance on BipedalWalker robustness tests during training (mean ± standard error), evaluated every 1k PPO updates.

*Table 2.* Generalization test returns and solved rates at 20k updates (mean ± standard error) for 5 runs of each method on BipedalWalker. Bold values are within one standard error of the best mean.

| ALGORITHMS | TEST RETURNS | SOLVED RATES |
|---|---|---|
| DR | $9.94 \pm 5.75$ | $0.01 \pm 0.00$ |
| PLR | $15.39 \pm 2.84$ | $0.02 \pm 0.00$ |
| ACCEL | $18.14 \pm 5.31$ | $0.04 \pm 0.00$ |
| PATH | $\mathbf{42.95 \pm 1.31}$ | $\mathbf{0.11 \pm 0.01}$ |

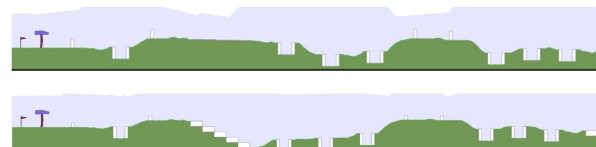

*Figure 9.* Examples of BipedalWalker terrains from the generalization test that PATH solves.

icy, PATH achieves substantially higher average test returns ($\geq 40$) and solved rates ($\geq 10\%$) than all baselines. To further characterize where each method succeeds, we collect the set of test configurations it solves and visualize them in Figure 8. We see PATH exhibits markedly broader and denser coverage of the environment space. Beyond better generalization, PATH locates the policy's learning frontier in $\Theta$, thereby identifying the region that is learnable in practice. Figure 9 shows a representative test environment that PATH solves while other methods fail after 20k updates. Together, these results indicate that PATH is more generalized and more robust than competing approaches. Overall, Figures 4 and 6 show clear advantages for methods that exploit the curriculum graph structure (PATH, AC-CEL) over baselines that do not (PLR, DR). This supports the prerequisite property in Section 2.2: leveraging curriculum structure yields a consistent training benefit. PATH further amplifies this advantage by actively refining path selection over the graph, outperforming ACCEL.

### 4.3. Random Exploration vs. Active Path Allocation

For BipedalWalker we observe that PATH switches at 8k updates when a sufficient number of sampled paths ended. To disentangle the effect of this phase transition, we take

the 8k checkpoint of PATH and continue training it to 20k while keeping applying PATH:RANDOM. In addition, we include a PATH:ACTIVE baseline trained for the full 20k updates. This setup enables a comprehensive comparison among PATH with its two components.

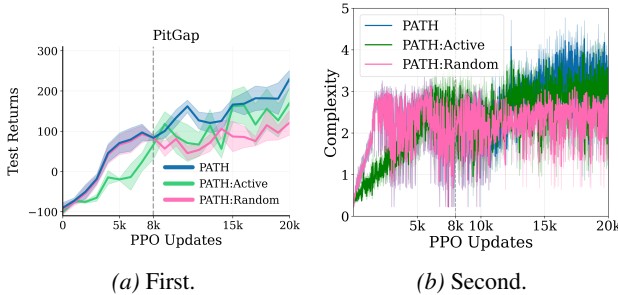

*(a)* First.  *(b)* Second.

*Figure 10.* Ablation results on `PitGap`. (a) Robustness test returns versus PPO updates for PATH, PATH:ACTIVE, and PATH:RANDOM (phase shift at ∼8k updates). (b) Corresponding sampling complexity over dimension of `PitGap` versus PPO updates for 3 methods. Both figures show the mean ± standard error. PATH overlaps with PATH:RANDOM in the first 8k updates.

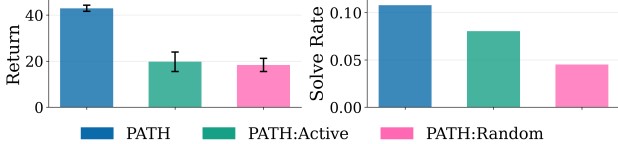

*Figure 11.* Bar figures of generalization test returns and solved rates at 20k updates (mean ± standard error) for 5 runs of PATH and its two components. bar height is the mean and segment shows one standard error.

We follow the robustness and generalization tests in Section 4.2. Robustness results are summarized in Figure 17 in Appendix C.4.1 and one case study is shown in Figure 10. In Figure 10a, PATH:RANDOM performs stronger early but then plateaus, whereas PATH:ACTIVE improves more slowly at first but continues improvements up to 20k updates, eventually overtaking PATH:RANDOM; PATH achieves superior performance in both start and end of training. Figure 10b visualizes the corresponding `PitGap` dimension of the parameters on sampled environments during training. PATH:RANDOM rapidly increases sampling complexity early, but later exhibits high repetition with lim-

ited further progress. In contrast, PATH:ACTIVE steadily increases sampling complexity throughout training, reaching harder environments by the end. PATH combines both behaviors: it explores quickly in the initial random phase and then reallocates sampling toward the most challenging environments, yielding consistently high sampling complexity and the strongest returns. Figure 10 offers a sufficient example that substantiates the discussion in Figure 2. We provide more direct evidence by evaluating generalization, a metric that more faithfully captures the degree of coverage. As shown in Figure 11, PATH exhibits a clear advantage over both PATH:RANDOM and PATH:ACTIVE, suggesting that the initial exploration followed by active path selection greatly improves generalization. Another advantage of combining the two phases is computational: although PATH:ACTIVE attains comparable robustness performance in the end, each update iteration takes nearly twice as long as in the random stage (Appendix B.3). Thus, PATH improves not only training effectiveness but also wall-clock efficiency, benefiting from the faster updates enabled by random path sampling. Collectively, our ablations suggest that PATH uses the training budget efficiently, achieving broader coverage of the environment space by leveraging both components.

**Remarks** We assume access to a curriculum prior and rely on hyperparameters such as the early-stop threshold $\varepsilon_{es}$. However, structured curricula are common in challenging RL and robotics tasks, where task difficulty is often increased via manual, multi-dimensional features' modifications (Margolis et al., 2022; Atanassov et al., 2024). Within this framework, early-stop serves as a lightweight mastery check for progression rather than a finely tuned control policy. Our ablations further show that PATH is robust to hyperparameter choices (Appendix C.4.2). We additionally verify PATH's robustness to noisy DAG structure in Appendix C.4.3.

## 5. Related Work

**Automated curriculum design for RL.** Curriculum learning accelerates RL by organizing experience into progressions of difficulty (Narvekar et al., 2020; Parker-Holder et al., 2022; Soviany et al., 2022). A broad literature automates this process via (i) teacher–student task selection driven by learning-progress/competence signals (Matiisen et al., 2017; Portelas et al., 2019), (ii) self-paced reweighting of the training distribution toward harder tasks as the agent improves (Klink et al., 2020; 2021), and (iii) goal generation curricula that expand outward from solvable goals to harder instances (Florensa et al., 2018; Colas et al., 2019). Some methods generate diverse tasks specifically to induce generalizable skill repertoires for downstream transfer (Fang et al., 2021). Complementary work improves

sim-to-real transfer by training over randomized environment distributions and focusing sampling on the most useful regions (Tobin et al., 2017; Peng et al., 2018; Muratore et al., 2021; Mehta et al., 2020), and by injecting additional structure or priors to guide progression (Tzannetos et al., 2024; Li et al., 2024; Wang et al., 2025; Hsiao et al., 2025). Among these autonomous algorithms, Svetlik et al. (2017) is closely related to our work in explicitly modeling curricula as a DAG. However, Svetlik et al. address a fundamentally different problem: given tasks with *unknown* prerequisite structure, they infer the curriculum DAG from pairwise task transfer matrices—their contribution answers *"what DAG to use?"* Our contribution answers the complementary question: *"given a DAG, how to traverse it efficiently?"* The two approaches are orthogonal and directly composable: Svetlik's DAG-inference procedure could construct the input to PATH for domains without explicit parameterization, creating a full pipeline from unstructured task sets to efficient curriculum traversal.

**Unsupervised Environment Design.** One prominent line of autonomous curriculum learning is Unsupervised Environment Design (UED) (Dennis et al., 2020), which casts training as a bilevel game over minimax regret objective. A complementary, replay-based thread approximates this objective by actively selecting and replaying environments from a buffer, with Prioritized Level Replay (PLR) (Jiang et al., 2021b) as a representative approach.

Recent work has sought to stabilize UED training through better-motivated objectives, estimators, and optimization procedures (Jiang et al., 2021a; Mediratta et al., 2023; Beukman et al., 2024; Rutherford et al., 2024; Monette et al., 2025). Another line improves robustness by encouraging smoother and safer alignment between the generator-induced environment distribution and a target distribution (Jiang et al., 2022; Azad et al., 2023; Costales & Nikolaidis, 2024; Lee et al., 2025). To improve the quality and diversity of the generated environments, some methods refine the regret/priority signals used for selection and replay (Li et al., 2023), while others incorporate explicit generative models to synthesize environment instances (Teoh et al., 2024; Chung et al., 2024). These works improve UED curricula by improving the generator—i.e., learning to produce higher-quality training environments—whereas we assume a fixed pool of environments and study how to learn effectively by sampling within this pool.

The most closely related work is ACCEL (Parker-Holder et al., 2022), which brings evolutionary ideas from POET (Wang et al., 2019) with PLR's prioritization, achieving SOTA performance among replay-based methods. In contrast to ACCEL and prior UED approaches that primarily improve how environments are generated or prioritized, our work focuses on exploiting curriculum structure itself: we make the implicit progression induced by evolutions ex-

plicit as a curriculum DAG, and introduce a path-level selection mechanism.

## 6. Conclusion

We introduced PATH, a new curriculum framework that operates active learning over the curriculum itself. We provide motivation for the two-phases learning benefiting from rapid exploration followed by regret based concentration. PATH assumes access to a constructible curriculum DAG, which is natural in parameterized or observable settings but requires additional effort in fully underspecified domains; automatic DAG construction (e.g., via LLMs as in Eurekaverse (Liang et al., 2024)) is a promising future direction. Our evaluation focuses on UED-family baselines; comparison with a broader set of curriculum learning approaches remains an open direction. Applying PATH to more complex real-world settings, including 3D embodied environments and sim-to-real transfer curricula, is an important next step.

## Impact Statement

This paper presents work whose goal is to advance the field of Machine Learning by improving curriculum learning for reinforcement learning via active path selection on structured curriculum graphs. The primary anticipated impact is methodological: better sample efficiency and stronger generalization may reduce the interaction and computation needed to train robust policies. At the same time, more efficient training methods can also accelerate the development and deployment of capable RL agents, which may be applied in domains with safety or misuse concerns. Our work does not introduce new sensing, surveillance, or data collection mechanisms, and we evaluate only in standard simulated benchmarks; nonetheless, responsible downstream deployment should include domain-specific safety constraints, testing, and monitoring.

## Acknowledgements

This work used Bridges-2 at Pittsburgh Supercomputing Center through allocation CIS240925 from the Advanced Cyberinfrastructure Coordination Ecosystem (Boerner et al., 2023): Services & Support (ACCESS) program, supported by the U.S. National Science Foundation (NSF) under grants 2138259, 2138286, 2138307, 2137603, and 2138296. We gratefully acknowledge the support of NSF IIS-2313131, IIS-2332475, IIS-2543755, the US Department of Energy award DE-EE0009505, the NSF-Simons AI-Institute for the Sky (SkAI) via grants NSF AST-2421845 and Simons Foundation MPS-AI00010513, and the University of Chicago's Research Computing Center for their support of this work.

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

## A. Detailed Algorithm of PATH

We provide the full pseudocode of PATH below (Algorithm 2). For clarity, we define several notations used in the algorithm: $\mathrm{Src}(\mathcal{G})$ denotes the set of source (root) nodes of the curriculum DAG $\mathcal{G}$, i.e., nodes with no incoming edges; $\mathrm{Sink}(\mathcal{G})$ denotes the set of sink (leaf) nodes with no outgoing edges; $\mathrm{Succ}_{\mathcal{G}}(\theta)$ denotes the set of immediate successors of node $\theta$ in $\mathcal{G}$; and $\mathrm{PLRScore}(\theta)$ computes the PLR positive value loss $S(\theta)$ as defined in Section 3.2.

---

**Algorithm 2** PATH: Random → Active

---

1: **Input:** curriculum DAG $\mathcal{C} = (\Theta, \mathcal{E})$ with source set $\mathrm{Src}(\mathcal{C})$, sink set $\mathrm{Sink}(\mathcal{C})$, and successor set $\mathrm{Succ}_{\mathcal{C}}(\theta)$; buffer size $N$; minibatch size $B$; early stop reward $\varepsilon_{\mathrm{es}}$; sampling patience $\varepsilon_{\mathrm{p}}$ (Random only); transition step $T_{\mathrm{switch}}$; replay rate $p$ (Active only); top-$k$ parameter $k$ (Active only); Regret score $S(\theta) = \mathrm{PLRSCORE}(\theta, \pi)$ (Active only).

2: **State:** policy $\pi$; stage $\in \{\mathrm{RANDOM}, \mathrm{ACTIVE}\}$; buffer entries $\{(b_i, w_i)\}_{i=1}^N$ with current level $b_i \in \Theta$ and weight $w_i > 0$; counters $\{c_i\}_{i=1}^N$ (Random only).

3: **Init:** stage $\leftarrow$ RANDOM; for $i \in [N]$: $b_i \sim \mathrm{Unif}(\mathrm{Src}(\mathcal{C}))$, $w_i \leftarrow 1$, $c_i \leftarrow 0$, $n \leftarrow 0$, $s \leftarrow 0$.

4: **while** not converged **do**

5:   **if** stage $=$ RANDOM **then**

6:     Sample indices $I \sim \mathrm{Unif}([N])^B$.

7:     **for** each $i \in I$ **do**

8:       $\theta \leftarrow b_i$; rollout on $\theta$; update $\pi$; compute average return $\widehat{J}(\theta)$; $c_i \leftarrow c_i + 1$.

9:       **if** $\widehat{J}(\theta) \geq \varepsilon_{\mathrm{es}}$ **or** $c_i \geq \varepsilon_{\mathrm{p}}$ **then**

10:        $b_i \leftarrow \begin{cases} \mathrm{Unif}(\mathrm{Src}(\mathcal{C})) & \theta \in \mathrm{Sink}(\mathcal{C}) \text{ or } c_i \geq \varepsilon_{\mathrm{p}} \\ \mathrm{Unif}(\mathrm{Succ}_{\mathcal{C}}(\theta)) & \text{otherwise} \end{cases}$

11:        $n \leftarrow n + \mathbb{I}\{\theta \in \mathrm{Sink}(\mathcal{C}) \text{ or } c_i \geq \varepsilon_{\mathrm{p}}\}$; $\quad c_i \leftarrow 0$; $\quad w_i \leftarrow 1$.

12:     $s \leftarrow s + 1$.

13:     **if** $s \bmod 1000 = 0$ and $n \geq T_{\mathrm{switch}}$ **then**

14:       stage $\leftarrow$ ACTIVE; set $w_i \leftarrow 1$ for all $i$ and discard counters $\{c_i\}$.

15:   **if** stage $=$ ACTIVE **then**

16:     **if** with probability $p$ **then**

17:       Update $p(i) = w_i / \sum_{j=1}^N w_j$; sample $I \sim p^B$; set $E \leftarrow \emptyset$.

18:       **for** each $i \in I$ **do**

19:         $\theta \leftarrow b_i$; rollout on $\theta$; update $\pi$; compute average return $\widehat{J}(\theta)$; set $w_i \leftarrow S(\theta)$.

20:         **if** $\widehat{J}(\theta) \geq \varepsilon_{\mathrm{es}}$ **then**

21:           $E \leftarrow E \cup \{(i, w_i)\}$.

22:       $m \leftarrow \min\{|E|, k\}$; $I_{\mathrm{top}} \leftarrow$ the $m$ indices in $E$ with largest weights; pad to $L$ of size $B$.

23:       **for** each $i \in L$ **do**

24:         $\theta \leftarrow b_i$; sample $\theta^+ \leftarrow \begin{cases} \mathrm{Unif}(\mathrm{Src}(\mathcal{C})) & \theta \in \mathrm{Sink}(\mathcal{C}) \\ \mathrm{Unif}(\mathrm{Succ}_{\mathcal{C}}(\theta)) & \text{otherwise} \end{cases}$; append $(\theta^+, 1)$ in the buffer.

25:     **else**

26:       Sample $\theta_1, \ldots, \theta_B \sim \mathrm{Unif}(\mathrm{Src}(\mathcal{C}))$; append $(\theta_t, 1)$ for all $t \in [B]$.

27:     While buffer size $> N$, delete a minimum-weight entry.

---

## B. Implementation Details

This section details (i) the environment configurations used in our experiments, (ii) the curriculum design and environment-generation procedure instantiated by each baseline (DR, PLR, ACCEL, and PATH), and (iii) training information and hyperparameters used for all methods. For PATH, we explicitly construct a curriculum DAG that matches the edit-based curriculum structure used implicitly by ACCEL; this alignment ensures comparisons isolate algorithmic differences in *how* the curriculum is exploited, rather than differences in the underlying curriculum itself.

## B.1. MiniGrid

**Environment**   We consider a partially observable navigation task (Dennis et al., 2020) on a $15 \times 15$ grid, implemented by Chevalier-Boisvert et al. (2018). Each cell is either a clutter (block), free space, the agent, or the goal. An episode terminates when the agent reaches the goal or when the time limit $T_{\max} = 250$ is reached. If the agent reaches the goal at time step $T$, it receives a terminal reward $1 - T/T_{\max}$; otherwise the reward is $0$. Observations are partial: the agent observes only a $5 \times 5$ egocentric window in front of it, and actions move the agent by one cell in the four cardinal directions at each time step. We consider an environment grid solved when the policy's return exceeds $0$.

**Curriculum**   We compare against DR, PLR, and ACCEL under a common layout space and clutters budget. Let $\Theta$ denote the set of all valid $15 \times 15$ grid layouts with at most $60$ interior clutters (excluding boundary walls), with a *fixed* agent start cell and a *fixed* goal cell chosen once at initialization and held constant thereafter. DR and PLR treat $\Theta$ as an unstructured set and sample layouts i.i.d.: they first sample a clutter count $w \sim \mathrm{Unif}\{0, \ldots, 60\}$ and then place $w$ clutters uniformly at random among interior cells, rejecting and resampling any placement that occupies the fixed start or goal. In contrast, ACCEL induces an implicit curriculum through *edit-based* layout evolution from an empty room. To make this curriculum explicit (and shared for fair comparison), we instantiate a curriculum DAG $\mathcal{G} = (\Theta, \mathcal{E})$ whose source nodes is the empty room with the random start/goal, and whose edges encode minimal topologically continuous changes: for $\theta, \theta' \in \Theta$, we include a directed edge $(\theta \to \theta') \in \mathcal{E}$ if and only if $\theta'$ is obtained from $\theta$ by a single valid local edit, namely inserting one interior clutter while preserving validity with the fixed start/goal. Denoting the out-neighborhood by $\mathcal{N}^+(\theta) = \{\theta' \in \Theta : (\theta \to \theta') \in \mathcal{E}\}$, exploration over this curriculum is a Markov process on $\mathcal{G}$: starting at the source (empty room), the next layout is drawn uniformly from the outgoing edges, i.e., $P(\theta' \mid \theta) = \mathbf{1}\{\theta' \in \mathcal{N}^+(\theta)\}/|\mathcal{N}^+(\theta)|$. Following ACCEL, we allow small stochastic edit distances by sampling $K \sim \mathrm{Unif}\{1, \ldots, 5\}$ and applying $K$ successive one-edit transitions, equivalently $\theta_{t+1} \sim P^{(K)}(\cdot \mid \theta_t)$. If $\mathcal{N}^+(\theta_t)$ is empty or the clutter number is up to the limit, the process restarts at the source node, ensuring sampled environments evolve via small, topologically continuous edits within the same layout space used by all baselines.

## B.2. BipedalWalker

**Environment**   We use a modified version by Parker-Holder et al. (2022) that enables active parameterization of the terrain complexity of BipedalWalker environment in OpenAI Gym (Brockman et al., 2016)). The agent observes a 24-dimensional proprioceptive state (lidar readings, joint angles, and contact indicators) without access to absolute position. Actions are four continuous motor torques. The agent receives a positive reward each step for forward progress and a penalty of $-100$ upon falling. Successfully traversing the terrain within the 2000 step horizon typically yields a total return exceeding 300. We consider a terrain instance *solved* if the policy achieves a return of at least 230.

*Table 3.* Environment parameter ranges and editing details of ACCEL on BipedalWalker

|  | STUMP HEIGHT | STAIR HEIGHT | STAIR STEPS | ROUGHNESS | PIT GAP |
|---|---|---|---|---|---|
| INITIATED VALUE | $[0, 0.4]$ | $[0, 0.4]$ | 1 | $\mathrm{Unif}(0, 0.6)$ | $[0, 0.8]$ |
| EDIT SIZE | 0.2 | 0.2 | 1 | $\mathrm{Unif}(0, 0.6)$ | 0.4 |
| MAX VALUE | $[5, 5]$ | $[5, 5]$ | 9 | 10 | $[10, 10]$ |

**Curriculum**   We follow the ACCEL configuration for BipedalWalker. Let $\mathcal{V} \subset \mathbb{R}^d$ ($d = 8$) denote the set of all valid terrain-parameter vectors $v \in \mathbb{R}^d$ under the coordinate-wise bounds $0 \leq v_i \leq v_i^{\max}$ (See MAX VALUE in Table 3), and let $s$ be a random seed. DR and PLR treat $\mathcal{V}$ as an unstructured design space and sample environment instances i.i.d. by drawing $v$ and $s$ directly. In contrast, ACCEL induces an implicit curriculum through *edit-based* evolution of $v$ starting from $v^{(0)}$ from the initialized parameter. To make this curriculum explicit, we instantiate a curriculum DAG $\mathcal{G} = (\Theta, \mathcal{E})$ whose nodes are environment instances $\theta = (v, s) \in \Theta := \mathcal{V} \times \mathcal{S}$, and whose edges encode minimal ACCEL mutations: for $\theta = (v, s)$ and $\theta' = (v', s')$, we include a directed edge $(\theta \to \theta') \in \mathcal{E}$ if and only if $\theta'$ is obtained from $\theta$ by a single valid coordinate edit, i.e., selecting an index $j \in \{1, \ldots, d\}$ and setting

$$v'_j = \min\{v_j + \Delta_j, \, (v_{\max})_j\}, \qquad v'_i = v_i \ \ (i \neq j),$$

where $\Delta_j$ is the fixed edit size for coordinate $j$ (Table 9). Denoting the out-neighborhood by $\mathcal{N}^+(\theta) = \{\theta' \in \Theta : (\theta \to \theta') \in \mathcal{E}\}$, exploration over this curriculum is a Markov process on $\mathcal{G}$: given the current node $\theta_t$, the next node is drawn uniformly from outgoing edges, i.e., $P(\theta' \mid \theta) = \mathbf{1}\{\theta' \in \mathcal{N}^+(\theta)\}/|\mathcal{N}^+(\theta)|$ with one random seed. Following ACCEL,

we allow small stochastic edit distances by sampling $K \sim \mathrm{Unif}\{1, 2, 3\}$ and applying $K$ successive one-edit transitions, equivalently $\theta_{t+1} \sim P^{(K)}(\cdot \mid \theta_t)$. Small edit sizes yield functional, local curriculum progress, while the fixed mutation scale introduces controlled variation that promotes generalization across nearby configurations.

### B.3. Training Information

For MiniGrid, each run (one seed per method) uses a single NVIDIA Tesla V100 GPU. Training for up to 20k PPO updates takes approximately 50 hours per seed for DR and PATH:RANDOM, and approximately 100 hours per seed for PLR and ACCEL.

For BipedalWalker, each run uses a single NVIDIA A40 GPU. Training for up to 20k PPO updates takes approximately 60 hours per seed for DR and PATH:RANDOM, 100 hours per seed for PATH (transitioning from PATH:RANDOM to PATH:ACTIVE at 8k updates), and approximately 160 hours per seed for PLR, ACCEL, PATH:ACTIVE.

### B.4. Hyperparameters

We inherit identical hyperparameters for PPO training and baselines (DR,PLR,ACCEL) configurations from (Parker-Holder et al., 2022). We conclude them with PATH's hyperparameters in Table 9.

## C. Full Experiments Details

In this section, we detail the construction of the MiniGrid and BipedalWalker test datasets and report the corresponding results referenced in Section 4.

### C.1. MiniGrid

#### C.1.1. ROBUSTNESS TEST

We use the fixed, challenging MiniGrid evaluation suite introduced by Jiang et al. (2021a), shown in Figure 12. For each layout, the agent's start cell, goal cell, and initial orientation are held fixed across all evaluation episodes. For each method, we evaluate the *nondeterministic* policy 20k PPO updates over 100 episodes and report the corresponding solved rate. Detailed results are provided in Table 4. Overall, PATH is both stable and consistently outperforms the baselines, achieving perfect solved rates on most test grids.

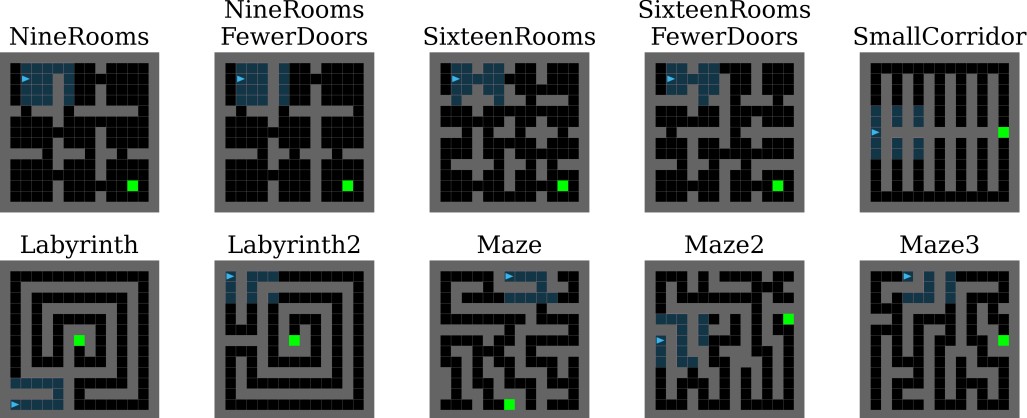

*Figure 12.* Specific layouts for each MiniGrid environment in the robustness test set.

*Table 4.* Solved rates after 20k PPO updates (mean ± standard error) over 5 independent runs for each method on each fixed MiniGrid layout (robustness test). For each run, we evaluate the final policy for 100 episodes per layout. Bold entries are within one standard error of the best mean.

|  | DR | PLR | ACCEL | PATH |
|---|---|---|---|---|
| NINEROOMS | **1.00 ± 0.00** | 0.88 ± 0.12 | **1.00 ± 0.00** | **1.00 ± 0.00** |
| NINEROOMSFEWERDOORS | 0.58 ± 0.17 | 0.99 ± 0.01 | 0.88 ± 0.12 | **1.00 ± 0.00** |
| SIXTEENROOMS | 0.84 ± 0.16 | 0.98 ± 0.02 | 0.98 ± 0.02 | **1.00 ± 0.00** |
| SIXTEENROOMSFEWERDOORS | 0.42 ± 0.18 | 0.61 ± 0.18 | 0.28 ± 0.19 | **0.87 ± 0.13** |
| SMALLCORRIDOR | **1.00 ± 0.00** | 0.99 ± 0.01 | **1.00 ± 0.00** | **1.00 ± 0.00** |
| LABYRINTH | 0.60 ± 0.24 | **1.00 ± 0.00** | 0.58 ± 0.22 | **1.00 ± 0.00** |
| LABYRINTH2 | 0.82 ± 0.18 | 0.55 ± 0.23 | **0.99 ± 0.01** | 0.94 ± 0.06 |
| MAZE | 0.33 ± 0.19 | 0.60 ± 0.24 | 0.79 ± 0.19 | **1.00 ± 0.00** |
| MAZE2 | 0.23 ± 0.14 | 0.59 ± 0.19 | **1.00 ± 0.00** | **1.00 ± 0.00** |
| MAZE3 | 0.80 ± 0.19 | 0.75 ± 0.15 | 0.86 ± 0.11 | **0.99 ± 0.01** |
| MEAN | 0.66 ± 0.09 | 0.79 ± 0.06 | 0.84 ± 0.07 | **0.98 ± 0.01** |

### C.1.2. GENERALIZATION TEST

To assess policy generalization, we construct a held-out test set by uniformly sampling 1000 valid grid layouts from the environment space $\Theta$. Figure 13 shows that these test grids are broadly distributed across the space under the two complexity dimensions. For evaluation, each method's trained policy is executed deterministically on every test grid for a single episode, and we report the overall solve rate across the full test set. The results are summarized in Table 1: PATH solves nearly all test environments, indicating strong generalization.

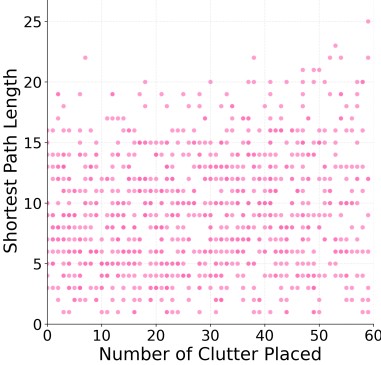

*Figure 13.* Distribution of 1000 valid grids uniformly sampled from $\Theta$ (clutter vs. shortest-path length).

## C.2. Bipedalwalker

### C.2.1. ROBUSTNESS TEST

To evaluate robustness on BipedalWalker, we construct a test suite comprising six environment classes (examples in Figure 14). `Basic` and `HardCore` are the standard OpenAI Gym environments (Brockman et al., 2016). The remaining four classes follow the fixed ACCEL configurations of Parker-Holder et al. (2022): `Stairs` uses stair height $[2, 2]$ with 5 steps; `PitGap` fixes the pit gap to $[5, 5]$; `Stump` fixes stump height to $[2, 2]$; and `Roughness` sets ground roughness to 5. For each class, we generate 128 terrains using distinct seeds and evaluate the policy *deterministically* for one episode per terrain to obtain returns. Detailed results are reported in Table 5.

*Table 5.* Average returns after 20k PPO updates (mean ± standard error) over 5 independent runs for each method on each robustness test class of BipedalWalker. For each run, we evaluate the final policy for 1 episodes for 128 terrains of each class. Bold entries are within one standard error of the best mean.

|  | DR | PLR | ACCEL | PATH |
|---|---|---|---|---|
| BASICS | $265.2 \pm 27.2$ | $311.5 \pm 2.2$ | $312.5 \pm 4.9$ | $\mathbf{321.6 \pm 1.1}$ |
| HARDCORES | $31.5 \pm 10.1$ | $81.8 \pm 11.0$ | $183.6 \pm 10.1$ | $\mathbf{240.5 \pm 9.0}$ |
| ROUGHNESS | $66.8 \pm 20.9$ | $123.5 \pm 16.1$ | $230.6 \pm 14.5$ | $\mathbf{266.9 \pm 5.2}$ |
| PITGAP | $21.5 \pm 8.1$ | $38.1 \pm 7.3$ | $39.1 \pm 12.0$ | $\mathbf{229.6 \pm 21.8}$ |
| STAIRS | $37.5 \pm 7.6$ | $44.2 \pm 5.1$ | $53.7 \pm 11.9$ | $\mathbf{177.6 \pm 23.7}$ |
| STUMP | $-19.9 \pm 8.0$ | $19.5 \pm 8.0$ | $93.4 \pm 31.7$ | $\mathbf{238.9 \pm 15.7}$ |
| MEAN | $67.1 \pm 41.3$ | $103.1 \pm 44.3$ | $152.2 \pm 44.3$ | $\mathbf{245.8 \pm 19.3}$ |

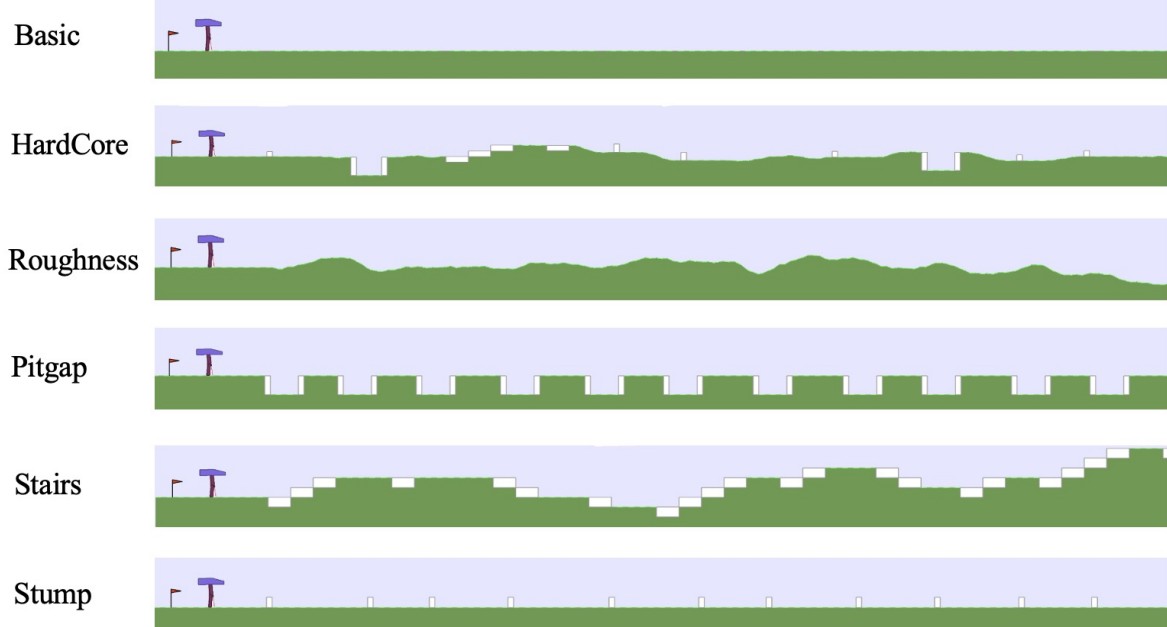

*Figure 14.* Examples of BipedalWalker terrain classes used for robustness testing.

### C.2.2. GENERALIZATION TEST

We sample 1000 terrains from the BipedalWalker environment space $\Theta$ by drawing terrain-parameter vectors at random and pairing each draw with a random seed, forming a held-out generalization test set (examples in Figure 15). For this dataset, we sample parameters directly from their continuous ranges (without one input of intervals), so each terrain is specified by an exact 5-dimensional parameter vector. We evaluate each method by running the learned policy *deterministically* for one episode on every test terrain, and report aggregate statistics in Table 2. Overall, this test set contains many highly challenging, often unsolved terrains, whereas PATH consistently pushes the learning frontier, achieving higher mean return and solving a larger fraction of terrains than the baselines.

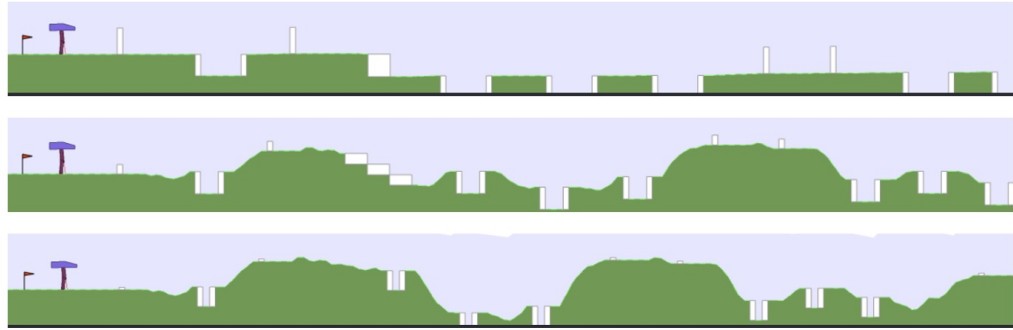

*Figure 15.* Examples from the BipedalWalker generalization test set of 1000 randomly sampled terrains.

## C.3. Leaper

### C.3.1. ENVIRONMENT DESCRIPTION

Leaper 16 is a Procgen game (Cobbe et al., 2020) in which a frog agent must cross a road (dodging cars) and a river (hopping on logs) to reach a goal lane. The state space is represented by $64 \times 64 \times 3$ pixel observations, and the action space follows the 15-dimensional Procgen game setting. It is a sparse-reward problem: the agent receives a reward of 0 or 10 upon success. Unlike MiniGrid or BipedalWalker, Leaper does not expose an explicit editing structure or difficulty parameter. Instead, difficulty emerges naturally from four observable environmental axes: (1) maximum car speed, (2) maximum log speed, (3) number of road lanes, and (4) number of water lanes.

This benchmark directly addresses the implicit parameterization setting described in Section 1(iii)—one that is more representative of realistic scenarios where environment parameters are not explicitly accessible.

### C.3.2. APPLYING PATH

We assume that increasing each axis leads to a harder environment, providing a natural difficulty ordering. We treat each axis as a curriculum dimension and assume monotone difficulty along it. This is how PATH is applied in the implicit case: by identifying observable structure and using it as a reasonable prior, without requiring a strictly guaranteed monotone DAG.

We represent the curriculum DAG using difficulty buckets, where each bucket is a 4-dimensional tuple. We randomly sample 300 Leaper environments as the training set. To construct buckets, we leverage CProcgen (Tan et al., 2023), a C++ Procgen backend that exposes the context parameters of each Procgen environment, allowing us to directly read out the four difficulty axes. We emphasize that this step can in principle be performed without explicit parameterization— observable environment statistics suffice. We compute statistics for each dimension, then discretize each axis into 3 levels (easy/medium/hard) with an approximately equal number of environments per level, yielding $3^4 = 81$ buckets. For instance, the number of car lanes ranges from 1 to 3 across the three difficulty levels along that dimension. The induced DAG starts at bucket $[0, 0, 0, 0]$; once PATH's early-stop criterion is satisfied, it randomly selects a neighboring bucket with one dimension incremented and samples uniformly from the environments within it.

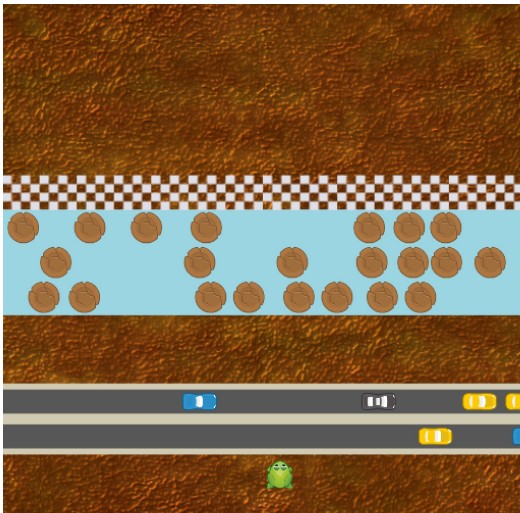

*Figure 16.* An example of the Leaper environment from Procgen.

### C.3.3. EXPERIMENT CONFIGURATION AND RESULTS

We use 300 training environments across 81 buckets, 32 parallel workers for PPO, 1000 total PPO update steps, rollout length 256, replay buffer size 30, early-stop threshold 10.0, patience 20 updates, and a phase shift after 500 PPO updates for PATH. We adopt the PPO hyperparameter settings from the PLR paper. We compare PATH, ACCEL, PLR, and DR across 5 seeds.

*Table 6.* Leaper generalization results. Evaluated on 100 held-out test environments, averaged over 5 seeds.

| Method | Solved Rate |
|--------|-------------|
| DR | $0.57 \pm 0.13$ |
| PLR | $0.45 \pm 0.08$ |
| ACCEL | $0.51 \pm 0.16$ |
| **PATH** | $\mathbf{0.66 \pm 0.04}$ |

PATH achieves the highest average solved rate with the lowest standard error across all methods. One notable observation is that DR is a strong baseline in this setting, likely due to the relatively small training set (300 environments), where uniform sampling already provides substantial coverage. However, PATH still outperforms DR, demonstrating that even when the curriculum DAG is not explicitly given but is instead constructed from observable difficulty axes, PATH's structured traversal yields meaningful gains over unstructured and replay-only methods.

### C.4. Ablations

We conduct several ablation experiments to provide a more comprehensive study of PATH.

### C.4.1. COMPONENT ABLATION: PATH:RANDOM VS. PATH:ACTIVE.

Our first ablation disentangles the contributions of PATH:RANDOM and PATH:ACTIVE. We report training curves on the robustness suite in Figure 17, and generalization performance in Figure 11. The analysis in Section 4.3 show that PATH benefits from *both* phases: PATH:RANDOM improves broad curriculum coverage, while PATH:ACTIVE further improves sample efficiency by concentrating training on environments with remaining learning potential.

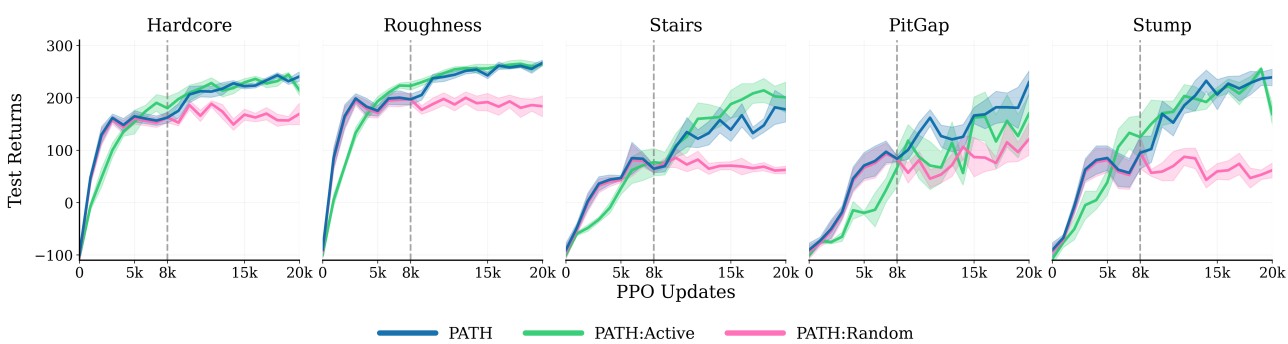

*Figure 17.* Performance on BipedalWalker robustness test environments during training (mean ± standard error), evaluated every 1k PPO updates. We compare three PATH variants: PATH runs PATH:RANDOM until 8k updates and then switches to PATH:ACTIVE; PATH:RANDOM keeps the random phase throughout training; PATH:ACTIVE uses the active phase from the beginning. The vertical dashed line marks the 8k transition point for PATH.

### C.4.2. HYPERPARAMETER ABLATION: EARLY STOPPING AND SAMPLING PATIENCE.

Our second ablation studies sensitivity to two hyperparameters: the early stop reward threshold $\mathcal{E}_{es}$ and the sampling patience $\mathcal{E}_p$. We denote a PATH instantiation by $\text{PATH}_{\mathcal{E}_{es}}^{\mathcal{E}_p}$. The default configuration used in this ablation comparisons corresponds to $\text{PATH}_{150}^{20}$. We compare it against $\text{PATH}_{150}^{30}$ (larger patience), $\text{PATH}_{100}^{20}$ and $\text{PATH}_{200}^{20}$ (weaker/stronger early stop thresholds), and $\text{PATH}_{\text{None}}^{20}$ (no early stopping). In $\text{PATH}_{\text{None}}^{20}$, each curriculum node is sampled exactly $\mathcal{E}_p = 20$ times before we proceed, regardless of observed returns.

For fair comparison, all methods use the same PATH:RANDOM phase for the first 8k PPO updates, then transfer to PATH:ACTIVE and continue training until 20k PPO updates. We report robustness results in Table 7 and generalization results in Table 8. The main observation is that $\text{PATH}_{\text{None}}^{20}$ performs substantially worse on both robustness and generalization. This is consistent with the role of early stopping in PATH: without it, a large fraction of the training budget is spent repeatedly revisiting nodes that are already effectively solved, which reduces coverage of the curriculum graph and slows progress toward unmastered regions.

In contrast, other variants—$\text{PATH}_{200}^{20}$, $\text{PATH}_{150}^{20}$, $\text{PATH}_{200}^{20}$, and $\text{PATH}_{150}^{30}$—achieve broadly similar performance, with the default $\text{PATH}_{150}^{20}$ attaining the best overall aggregate score in our study. This indicates an inherent trade-off in selecting $\mathcal{E}_{es}$, as discussed in Section 3.1. Increasing patience from $\mathcal{E}_p = 20$ to $\mathcal{E}_p = 30$ (i.e., $\text{PATH}_{150}^{30}$) is slightly worse than the default, supporting our design choice that $\mathcal{E}_p = 20$ is an effective upper bound for maintaining smooth curriculum progression without overspending on mastered nodes.

Overall, aside from the no early stop setting, PATH is relatively insensitive to reasonable choices of $\mathcal{E}_{es}$ and $\mathcal{E}_p$, and these configurations remain competitive and consistently stronger than baselines such as ACCEL. This ablation therefore supports PATH as a stable and effective curriculum learning method.

*Table 7.* Average returns after 20k PPO updates (mean ± standard error) across 5 independent runs for each ablation variant on the BipedalWalker robustness test classes. For each run, we evaluate the final policy for one episode on each of 128 terrains per class. Bold entries indicate results within one standard error of the best mean. $\text{PATH}_{150}^{20}$ is the default algorithm.

|  | $\text{PATH}_{150}^{20}$ | $\text{PATH}_{150}^{30}$ | $\text{PATH}_{100}^{20}$ | $\text{PATH}_{200}^{20}$ | $\text{PATH}_{\text{None}}^{20}$ |
|---|---|---|---|---|---|
| BASICS | $321.6 \pm 1.1$ | $320.4 \pm 2.2$ | $318.9 \pm 2.5$ | $\mathbf{322.4 \pm 2.6}$ | $312.6 \pm 2.4$ |
| HARDCORES | $240.5 \pm 9.0$ | $\mathbf{252.1 \pm 5.0}$ | $220.9 \pm 10.1$ | $226.2 \pm 7.8$ | $229.2 \pm 10.2$ |
| ROUGHNESS | $\mathbf{266.9 \pm 5.2}$ | $253.7 \pm 12.7$ | $241.9 \pm 7.4$ | $257.5 \pm 2.6$ | $245.5 \pm 5.7$ |
| PITGAP | $\mathbf{229.6 \pm 21.8}$ | $188.1 \pm 28.2$ | $164.9 \pm 29.8$ | $188.0 \pm 27.9$ | $167.1 \pm 45.7$ |
| STAIRS | $177.6 \pm 23.7$ | $192.8 \pm 20.6$ | $212.8 \pm 15.9$ | $\mathbf{203.8 \pm 25.5}$ | $189.1 \pm 28.8$ |
| STUMP | $\mathbf{238.9 \pm 15.7}$ | $238.0 \pm 12.0$ | $210.0 \pm 38.4$ | $220.4 \pm 16.2$ | $157.2 \pm 25.8$ |
| MEAN | $\mathbf{245.8 \pm 19.3}$ | $240.5 \pm 20.0$ | $228.2 \pm 21.0$ | $236.4 \pm 20.0$ | $216.8 \pm 24.0$ |

*Table 8.* Generalization test returns and solved rates at 20k updates (mean $\pm$ standard error) for 5 runs of each ablation variant on BipedalWalker. Bold values are within one standard error of the best mean. $\text{PATH}^{20}_{150}$ is the default algorithm.

| METRIC | $\text{PATH}^{20}_{150}$ | $\text{PATH}^{30}_{150}$ | $\text{PATH}^{20}_{100}$ | $\text{PATH}^{20}_{200}$ | $\text{PATH}^{20}_{\text{None}}$ |
|---|---|---|---|---|---|
| TEST RETURNS | $\mathbf{42.95 \pm 1.31}$ | $34.20 \pm 7.68$ | $25.80 \pm 6.86$ | $31.88 \pm 4.99$ | $-6.44 \pm 5.27$ |
| SOLVED RATE | $\mathbf{0.11 \pm 0.01}$ | $0.10 \pm 0.01$ | $0.09 \pm 0.01$ | $0.09 \pm 0.01$ | $0.07 \pm 0.00$ |

### C.4.3. ROBUSTNESS TO DAG MISSPECIFICATION

To directly evaluate PATH's robustness to noisy or incorrect curriculum edges, we introduce a *perturbation probability* $p_{\text{perturb}}$: at each successor selection after the early-stop criterion is met, with probability $p_{\text{perturb}}$ a uniformly random node from the full graph space is selected instead of the curriculum successor, explicitly breaking DAG structure. All results are averaged over 5 seeds. PATH shifts phase at 5k steps; total training is 10k updates.

*Table 10.* MiniGrid — Robustness Solved Rate at 10k (same setting as Table 4).

| Method | Robustness Solved Rate |
|---|---|
| DR | $0.18 \pm 0.1$ |
| PLR | $0.34 \pm 0.0$ |
| ACCEL | $0.56 \pm 0.1$ |
| PATH (50% perturb) | $0.43 \pm 0.1$ |
| PATH (30% perturb) | $0.53 \pm 0.1$ |
| PATH (10% perturb) | $0.67 \pm 0.1$ |
| **PATH** | $\mathbf{0.76 \pm 0.0}$ |

*Table 11.* BipedalWalker — Mean Robustness Return at 10k (same setting as Table 5).

| Method | Mean Robustness Return |
|---|---|
| DR | $51.60 \pm 10.4$ |
| PLR | $69.47 \pm 4.7$ |
| ACCEL | $142.31 \pm 9.7$ |
| PATH (50% perturb) | $139.06 \pm 12.3$ |
| PATH (30% perturb) | $168.24 \pm 8.1$ |
| PATH (10% perturb) | $184.63 \pm 7.3$ |
| **PATH** | $\mathbf{198.12 \pm 9.7}$ |

At 30% perturbation, PATH remains comparable to ACCEL on MiniGrid (0.53 vs. 0.56, within one standard error) and exceeds ACCEL on BipedalWalker (168.24 vs. 142.31). At 50% perturbation—half of all curriculum edges replaced with random jumps—PATH still substantially outperforms DR and PLR on both benchmarks. Performance degrades gradually rather than catastrophically, consistent with the two-level safeguard: $\varepsilon_{\text{es}}$ advances cheaply when a successor is unexpectedly easy, and $\varepsilon_p$ bounds budget waste when a successor provides no transfer benefit, reinitializing the pointer after at most 20–25 steps (Table 9). In the adversarial limit where every edge is uninformative, PATH degrades to PATH:RANDOM, which already substantially outperforms all baselines.

### C.4.4. TWO-PHASE VALIDATION ON MINIGRID

In the main experiments (Section 4), PATH never switches from PATH:RANDOM to PATH:ACTIVE within 20k updates on MiniGrid because near-ceiling performance (99.4% generalization solved rate) is achieved entirely during the random phase. To directly validate that the two-stage design provides value on MiniGrid when the random phase does not suffice, we run a shortened experiment: 10k total updates with phase shift at 5k, so that both phases must operate within the training budget.

*Table 12.* MiniGrid — Robustness Solved Rate under shortened horizon (5 seeds).

| Method | 5k updates | 10k updates |
|---|---|---|
| DR | $0.17 \pm 0.1$ | $0.18 \pm 0.1$ |
| PLR | $0.19 \pm 0.1$ | $0.38 \pm 0.0$ |
| ACCEL | $0.30 \pm 0.1$ | $0.56 \pm 0.1$ |
| PATH:ACTIVE | $0.33 \pm 0.1$ | $0.59 \pm 0.1$ |
| PATH:RANDOM | $0.35 \pm 0.0$ | $0.70 \pm 0.1$ |
| **PATH** | $\mathbf{0.35 \pm 0.0}$ | $\mathbf{0.76 \pm 0.0}$ |

At 5k updates (still in the random phase), PATH matches PATH:RANDOM exactly and already outperforms ACCEL and PATH:ACTIVE. By 10k updates, PATH (0.76) clearly outperforms both PATH:RANDOM (0.70) and PATH:ACTIVE (0.59), confirming that PATH:ACTIVE adds measurable value once the random phase saturates. The original 20k experiment simply did not require switching because near-ceiling performance was already achieved.

### C.4.5. BIPEDALWALKER GENERALIZATION IN THE SOLVABLE SUBSPACE

The full BipedalWalker parameter space $[10, 10, 5, 5, 9]$ contains a significant portion of inherently unsolvable configurations. To better isolate meaningful differences, we restrict the upper parameter limits to $[6, 6, 3, 3, 6]$—a more tractable subspace—and evaluate on 1000 uniformly sampled environments at 20k updates, averaged over 5 seeds.

*Table 13.* BipedalWalker generalization in the cropped solvable subspace.

| Method | Mean Score | Solved Rate |
|---|---|---|
| DR | $42.22 \pm 10.6$ | $0.043 \pm 0.02$ |
| PLR | $66.65 \pm 8.2$ | $0.096 \pm 0.02$ |
| ACCEL | $121.16 \pm 9.7$ | $0.277 \pm 0.03$ |
| **PATH** | $\mathbf{201.14 \pm 6.5}$ | $\mathbf{0.545 \pm 0.02}$ |

PATH achieves a 97% improvement in solved rate and 66% improvement in mean score over ACCEL, confirming that the relative gains observed in the full parameter space are large and significant when restricted to the solvable subspace.

*Table 9.* Hyperparameters for RL training and curriculum algorithms.

| Parameter | MiniGrid | BipedalWalker |
|---|---|---|
| *PPO* | | |
| $\gamma$ | 0.995 | 0.99 |
| $\lambda_{\mathrm{GAE}}$ | 0.95 | 0.9 |
| PPO rollout length | 256 | 2000 |
| PPO epochs | 5 | 5 |
| PPO minibatches per epoch | 1 | 32 |
| PPO clip range | 0.2 | 0.2 |
| PPO number of workers | 32 | 16 |
| Adam learning rate | 1e−4 | 3e−4 |
| Adam $\epsilon$ | 1e−5 | 1e−5 |
| PPO max gradient norm | 0.5 | 0.5 |
| PPO value clipping | yes | no |
| return normalization | no | yes |
| value loss coefficient | 0.5 | 0.5 |
| student entropy coefficient | 0.0 | 1e−3 |
| generator entropy coefficient | 0.0 | 0.0 |
| *PATH* | | |
| Early stop reward, $\varepsilon_{\mathrm{es}}$ | 0.7 | 150 |
| Sampling patience, $\varepsilon_{\mathrm{p}}$ | 25 | 20 |
| Transition step, $T_{\mathrm{switch}}$ | 200 | 200 |
| Replay rate, $p$ | 0.8 | 0.9 |
| Buffer size, $K$ | 4000 | 1000 |
| Scoring function | positive value loss | positive value loss |
| Temperature, $\beta$ | 0.3 | 0.1 |
| Staleness coefficient, $\rho$ | 0.3 | 0.5 |
| *ACCEL* | | |
| Edit rate, $q$ | 1.0 | 1.0 |
| Replay rate, $p$ | 0.8 | 0.9 |
| Buffer size, $K$ | 4000 | 1000 |
| Scoring function | positive value loss | positive value loss |
| Edit method | random | random |
| Number of edits | 5 | 3 |
| Prioritization | rank | rank |
| Temperature, $\beta$ | 0.3 | 0.1 |
| Staleness coefficient, $\rho$ | 0.3 | 0.5 |
| *PLR* | | |
| Scoring function | positive value loss | positive value loss |
| Replay rate, $p$ | 0.5 | 0.5 |
| Buffer size, $K$ | 4000 | 1000 |

