# OpenReview forum: "Active Curriculum Refinement for Reinforcement Learning"
_ICML.cc/2026/Conference — ICML 2026 regular_

### Official Review · Reviewer_Aadw · 2026-03-12

**Soundness:** 3
**Presentation:** 2
**Significance:** 2
**Originality:** 3
**Overall Recommendation:** 3
**Confidence:** 5

**Summary:**

This paper proposes a formalization of curriculum learning for RL as an active path acquisition problem on a directed acyclic graph (DAG) over environment instances. Then, the authors develop a method called PATH that consists of two stages: PATH:Random explores paths randomly to achieve broad coverage, and PATH:Active replays under-mastered nodes with high learning potential. The paper investigates empirical validation through experiments in MiniGrid and Bipedalwalker, where the former is a sparse-reward environment with a discrete task space, and the latter is a dense-reward environment with a continuous task space. Empirical results show improved robustness and generalization, while ablations indicate that two stages operate in a complementary fashion.

**Compliance With Llm Reviewing Policy:**

Affirmed.

**Final Justification:**

I thank the authors for their engagement throughout the rebuttal period.

The clarification, discussion, limitations, and perturbation experiments they have provided certainly strengthened their work and its presentation. However, I'll keep my recommendation as a weak reject, as I found their responses and additional work to address the limited benchmarks/baselines insufficient. There are two reasons behind this decision:
1) The Leaper environment, which they have experimented with during the rebuttal period, is still quite explicitly parameterized. The environment is configured along four observable dimensions. Even then, PATH doesn't clearly outperform baselines or supposedly SOTA methods.
2) The baselines are UED algorithms except for one curriculum learning method, which is quite old at this point. Without a comparison against recent approaches (UED and non-UED), which there are many, especially for explicitly parameterized environments such as those studied in this paper, it is not possible to assess the benefits of this algorithm.

I'm also increasing my confidence to 5. Although I believe this is a promising work, it needs more empirical support to substantiate its claims.

**Key Questions For Authors:**

- How sensitive is PATH to DAG quality, as the paper assumes that curriculum DAGs are given?
- Can authors clarify the notation inconsistencies mentioned in the weaknesses section?
- Is the method applicable to settings where the task is completely underspecified, e.g., it is identified by an index only?

**Limitations:**

Although the authors discuss some hyperparameters, they do not include a specific limitations section. In fact, even though it is not a requirement, I couldn't find the word "limitation" when I searched for it. Not only should they add a (sub)section on this, but they should also discuss the assumption on an a priori available curriculum DAG. There is also no discussion of future work.

**Strengths And Weaknesses:**

Strengths:
- The domains in the experimental setup have complementary properties, allowing evaluation of PATH’s effectiveness under sparse/dense rewards as well as in discrete/continuous task spaces. Ablations disentangle two stages and enable confirmation of the authors’ hypotheses regarding their use, as described in earlier sections. The authors answer multiple questions using different metrics, and the experiments involve sufficiently many independent training runs/seeds.

- Figure 1 is very effective at providing an overview of PATH’s two stages.  In general, Section 3 (especially Section 3.4) describes the authors’ intuition on random exploration for coverage, supported by active allocation to residual regions, as illustrated in Figure 2.

- Automated curriculum generation is a very practical problem for robotics. Gains in robustness and generalization are significant, especially in the dense-reward/continuous-task-space setting.

- Formalizing curriculum learning as active path acquisition on a DAG is a conceptual contribution, supported by a novel and well-principled two-stage algorithm, which makes use of some existing ideas, e.g., editing from ACCEL.

Weaknesses:
- Although the empirical validation addresses various questions, it examines only two domains. The paper focuses specifically on unsupervised environment design; hence, it lacks comparisons with other curriculum learning methods.

- There are some notable inconsistencies in notation: S($\theta$) is never linked to $\omega(\theta)$ in Section 3.2 and in Algorithm 1. A different symbol in the main text and the appendix indicates the replay rate. Some quantities in Algorithm 2 are introduced without proper context. In addition, the assumption that a DAG is given is confusing: what exactly is given?

- The applicability of the proposed algorithm in settings where the task space is truly underspecified, i.e., its parameterization is not given, is questionable. The proposed method relies on GAE for regret approximation, whereas GAE has been argued to be less reliable than positive value loss or maximum Monte Carlo.

- The formalization is presented as one of the main contributions; at the same time, Svetlik et al. (2017)’s work is mentioned as another DAG-based curriculum learning study. Therefore, a deeper discussion of both methods is needed.

---

> ### Author Rebuttal · Authors · 2026-03-28
>
> We thank Reviewer Aadw for the constructive feedback.
>
> ---
>
> ### W1 - limited benchmarks and baselines
>
> MiniGrid and Bipedalwalker follow established works such as ACCEL, ADD (Chung, H et al.) to span complementary RL regimes—sparse vs. dense reward, discrete vs. continuous control, and edit-based vs. parameter-based structures. The consistent gains across both domains suggest that PATH captures a general curriculum principle rather than a domain-specific heuristic. We agree that extending PATH to more realistic RL environments is an important future direction.
>
> We also believe the chosen baselines are appropriate for isolating why PATH works. In particular, Parker-holder et al. showed that ACCEL clearly outperforms prior automated curriculum methods without leveraging the curriculum structure of environments.
> This makes it a strong baseline for evaluating whether explicitly exploiting structure, as PATH does, provides additional benefit.
>
> ---
>
> ### W3,Q3 — GAE; case on underspecified task
>
> - We respectfully correct a misreading: in line 154 of Section 3.2 we are proposing the use of positive value loss but not GAE as an active learning strategy.
> - On applicability to underspecified tasks, in Section 1 we discuss this setting and cite Lee et al., 2025 as one example: when explicit parameters or edit operations are unavailable, feature embeddings can define edges via monotone progress along feature axes — e.g., a task difficulty classifier partitions index-only environments into difficulty buckets, and PATH can traverse the induced DAG over buckets.
>
> ### W2,Q2 — unclear notations
>
> We appreciate the careful reading and will revise the notation for clarity and consistency.
>
> - $S(\theta)$ and $w(\theta)$: $S(\theta)$ is the positive value loss computed from rollouts; $w(\theta)$ is the buffer weight, updated as $w(\theta) \leftarrow S(\theta)$ after each training step on $\theta$ (line 19 of Algorithm 2). We will add this explicitly in Algorithm 1.
> - Replay rate: We will unify to $p$ to represent the replay rate.
> - For quantities such as $\text{Src}(C)$, $\text{Sink}(C)$, and $\text{Succ}_C(\theta)$, we will add a paragraph at the beginning of Appendix A to formally define all symbols, with clear correspondence to graph definitions in Section 3.
> - The curriculum DAG encodes edges relationships between environment nodes. In MiniGrid, an edge $(\theta \to \theta’)$ exists iff $\theta’$ is obtained from $\theta$ via a single block-insertion edit (Appendix B.1); in BipedalWalker, iff $\theta’$ is obtained via a single coordinate increment (Appendix B.2). We will make this construction explicit in Algorithms.
>
> ---
>
> ### Q1 — DAG sensitivity
>
> We respectfully clarify that PATH does not require strict monotonic difficulty. In MiniGrid, $\texttt{NineRoomsFewerDoors}$ and $\texttt{SixteenRoomsFewerDoors}$ (Figure 12) contain fewer grids but are harder tasks with lower solved rates (Table 4). In BipedalWalker, uphill vs. downhill (uphill is generally harder) terrain is not strictly determined by parameters. Despite this, PATH consistently outperforms baselines, remaining robust to non-strictly monotonic DAGs by leveraging underlying curriculum structure.
>
> We further evaluate robustness of PATH via **perturbation**: each successor selection is replaced by a **random** node in the **full** graph space with the probability 10%/30%/50%. We conduct this experiment in both domains
>
> ---
>
> **MiniGrid — Robustness Solved Rate at 10k (same setting as Table 4)**
>
> |Method|Average Solved Rate|
> |---|---|
> |DR|0.177 ± 0.01|
> |PLR|0.379 ± 0.00|
> |ACCEL|0.557 ± 0.01|
> |PATH (50% perturb)|0.427 ± 0.01|
> |PATH (30% perturb)|0.532 ± 0.01|
> |PATH (10% perturb)|0.674 ± 0.01|
> |**PATH**|**0.762 ± 0.00**|
>
> **BipedalWalker — Mean Robustness Return at 10k (same setting as Table 5)**
>
> |Method|Average Return|
> |---|---|
> |DR|51.60 ± 10.40|
> |PLR|69.47 ± 4.68|
> |ACCEL|142.31 ± 9.71|
> |PATH (50% perturb)|139.06 ± 12.30|
> |PATH (30% perturb)|168.24 ± 8.12|
> |PATH (10% perturb)|184.63 ± 7.28|
> |**PATH**|**198.12** ± **9.69**|
>
> ---
>
> All results are averaged over 5 seeds, and PATH shifts phase at 5k steps. Performance degrades gradually rather than collapsing under perturbation. In MiniGrid, with 30% perturbation, PATH remains comparable to ACCEL; in BipedalWalker, even at 50%, PATH is comparable with ACCEL and substantially outperforms DR and PLR. Performance degrades gradually rather than catastrophically, demonstrating strong robustness to misspecified DAGs.
>
> ---
>
> ### W4 - Compare with Svetlik et al.'s work
>
> Svetlik et al., 2017 address a fundamentally different problem: given tasks with *unknown* prerequisite structure, infer the curriculum DAG from pairwise task transfer potential — their contribution answers "what DAG to use?" Our contribution answers "given a DAG, how to traverse it efficiently?" The two works are orthogonal and directly composable.
>
> We will incorporate the above clarifications, limitations, and future directions into the final version of the paper.

---

> > ### Author Rebuttal · Reviewer_Aadw · 2026-04-03
> >
> > Thank you for taking the time to respond to my review.
> >
> > W1 & Q3: In my review, I noted that the two domains studied in the paper are complementary. Yet Q3 raises another dimension that is quite different from the paper's domains of interest. Underspecified tasks are not trivial, and heuristics such as grouping IDs into "difficulty" buckets should be validated via experimentation. Such experimentation would also show whether PATH is applicable without explicit task parameterization.
> >
> > W3: Thank you for correcting my misreading.
> >
> > W2 & Q2: Thank you for the clarification. My concerns here are addressed.
> >
> > Q1: Thank you for these perturbation experiments. They give a good idea about how robust the method is to changes in the DAG. They fully address my concern. I'm not sure why you are talking about strict monotinicty in your response.
> >
> > W4: I believe this difference between Svetlik et al.'s work should be discussed in more detail in your paper.
> >
> > Given the initial rebuttal, I believe my concerns are partially resolved.
> > 1) I still believe that another domain, possibly one that is not explicitly parameterized, should have been studied, given that the paper compares only against UED methods.
> > 2) In addition, Parker-holder et al. compare ACCEL against only one non-UED curriculum learning method, namely, ALP-GMM, in their paper, and that paper is already four years old. So you cannot simply use the fact that ACCEL outperforms one SOTA method at a given time to argue against comparing against another method.
> > 3) I still do not see a discussion of limitations. Given that your paper doesn't even have the word "limitation", I believe this should have been addressed in your rebuttal already.
> >
> > Until my concerns are addressed, I will keep my score as is.

---

> > > ### Author Response · Authors · 2026-04-06
> > >
> > > We thank the reviewer Aadw for the continued engagement.
> > >
> > > ## Clarification
> > >
> > > We would like to clarify a potential source of confusion regarding "monotonicity" in the context of PATH. When we use this term, we are referring specifically to strict condition of **directed edges** in DAG — that is, every edge consistently points from an easier environment to a harder one. Our perturbation study provides empirical support for the robustness of this condition, suggesting that it can be relaxed; for example, not all edges need to follow an easy-to-hard sequence to facilitate training.
> > >
> > > We will include a dedicated paragraph in the final version clearly distinguishing our work from Svetlik et al. (2017).
> > >
> > > ## Concern 1 — Non explicitly parameterized domain.
> > >
> > > We agree that providing an experiment in this setting is important, as it complements the three main modes (Section 1) of applying PATH to generalized RL problems. We have conducted a new experiment on **Leaper** from the benchmark Procgen, a game where difficulty is not explicitly parameterized but emerges from observable environmental axes. **We refer the reviewer to our reply to the rebuttal acknowledgement by Reviewer 3WBd for the full environment description**. Briefly, we construct 81(4 dimension,3 difficulty level per dimension) buckets from 4 observable difficulty axes such as number of car lanes and speed of the car. We sample 300 training environments, and apply PATH with the DAG starting at the bucket [0,0,0,0]. Here is the result over 5 seeds:
> > >
> > > | Method | Solved Rate |
> > > |--------|-------------|
> > > | DR | 0.568 ± 0.130 |
> > > | PLR | 0.446 ± 0.082 |
> > > | ACCEL | 0.512 ± 0.162 |
> > > | PATH | 0.655 ± 0.042 |
> > >
> > > The result confirms that PATH outperforms other baselines. We observe that ACCEL, despite also using our defined DAG, does not achieve strong performance. This is a key point: simply defining or following the DAG is not sufficient for effective curriculum learning, which further highlights the value of PATH.
> > >
> > > This experiment demonstrates that PATH is applicable when explicit parameterization is unavailable, and that heuristic bucket construction from observable metrics suffices for meaningful curriculum gains.
> > >
> > > ## Concern 2 — Baselines
> > >
> > > We thank the reviewer for raising this important concern. ALP-GMM is indeed an older method, and we fully acknowledge this. What we intended to convey is that after 2020, the dominant trend in automated curriculum learning shifted decisively toward Unsupervised Environment Design (UED). Within this landscape, ACCEL — despite being four years old — remains one of the strongest and most widely used UED baselines: recent works such as CENIC [1] build orthogonally on ACCEL, using a Gaussian mixture model to increase the novelty of sampled environments, and still treat ACCEL as the primary reference point.  Crucially, PATH is deliberately lightweight — it relies purely on environment sampling and active learning scoring derived from rollouts only, without introducing any additional learned components. This contrasts with recent methods such as ADD [2], which augment UED with a diffusion model to generate harder environments. PATH's direct comparison to ACCEL therefore isolates the specific benefit of exploiting implicit curriculum structure. Combining these orthogonal innovations — structured traversal with learned generation or novelty estimation — is a promising future direction. We acknowledge the limitations of chosen baselines, as discussed below.
> > >
> > > ## Concern 3 — Limitations
> > >
> > > We agree that a dedicated Limitations section is essential and will add one to the final version of the paper. Drawing on the discussion with all reviewers, it will cover: (1) **DAG construction cost** — PATH assumes access to a constructible curriculum DAG, which is natural in parameterized or observable settings but requires additional effort in fully underspecified domains; automatic DAG construction (e.g., via LLMs as in Eurekaverse [3]) is a promising future direction. (2) **Breadth of baselines** — PATH currently focuses on UED-family methods; comparison with a broader set of curriculum learning approaches (e.g., teacher-guided or goal-conditioned methods) remains an open direction we intend to pursue. (3) **Real-world applicability** — while our experiments cover three complementary environment types, applying PATH to more complex, real-world settings remains an important open challenge and a key future direction.
> > >
> > > Finally, we would like to thank you again for your constructive feedback, which has greatly improved the quality of our paper.
> > >
> > >
> > > ---
> > > [1] Teoh, J., Li, W., and Varakantham, P. Improving environment novelty quantification for effective unsupervised environment design. NeurIPS 2024.
> > >
> > > [2] Chung, H., Lee, J., Kim, M., Kim, D., and Oh, S. Adversarial environment design via regret-guided diffusion models. NeurIPS 2024.
> > >
> > > [3] Liang, W. et al. Eurekaverse: Environment curriculum generation via large language models. Conference on Robot Learning, 2024.

---

### Official Review · Reviewer_H6PJ · 2026-03-13

**Soundness:** 3
**Presentation:** 4
**Significance:** 3
**Originality:** 4
**Overall Recommendation:** 4
**Confidence:** 3

**Summary:**

This paper formalizes curriculum learning as active path acquisition on a directed acyclic graph (DAG), where nodes are environment instances and edges encode prerequisite relations (e.g., adding obstacles, incrementing difficulty parameters). The proposed method PATH has two stages: PATH:RANDOM samples diverse maximal paths through the DAG to achieve broad coverage, advancing along a path when an early-stop reward threshold is met; PATH:ACTIVE then uses regret-based signals (positive value loss from PLR) to identify under-mastered regions and allocates additional training budget by dispatching multiple paths from high-regret nodes. Experiments on MiniGrid and BipedalWalker show PATH outperforms DR, PLR, and ACCEL on both robustness and generalization, with ablations confirming the complementary roles of the two phases.

**Compliance With Llm Reviewing Policy:**

Affirmed.

**Key Questions For Authors:**

In many practical domains, difficulty is not strictly increasing along a single axis, for instance, adding an obstacle in MiniGrid could make a layout easier if it blocks a deceptive dead-end. Does the early-stop mechanism handle such cases gracefully, or does PATH get stuck when a harder successor is actually easier than its predecessor? Relatedly, have you observed cases where the agent masters a child node before its parent?

In BipedalWalker, the DAG has 8 parameter dimensions with relatively small increments, yielding a moderate branching factor. What happens as the number of dimensions or the branching factor grows? does PATH:RANDOM still achieve meaningful coverage, or does the curse of dimensionality make random path sampling ineffective? Is there a regime where the DAG is too large for the two-phase strategy to work within a reasonable budget?

**Limitations:**

The method's applicability is fundamentally tied to the availability of a well-structured curriculum DAG with meaningful prerequisite relations. The paper demonstrates PATH only on environments where the DAG is trivially constructed from known parameterizations (add blocks, increment terrain parameters). For environments where difficulty is emergent, non-monotonic, or depends on complex interactions between features, constructing such a DAG requires significant domain expertise that may not be available. The paper does not discuss the cost of DAG specification, nor does it experiment with automatically discovering or learning the graph structure.

**Strengths And Weaknesses:**

**Strengths**

- Framing curriculum learning as active path acquisition on a DAG is a simple but effective conceptual contribution. The two-phase design is well-motivated: Section 3.4 provides clear intuition for why random exploration achieves broad coverage early (non-overlapping t-balls) but saturates, and why regret-driven allocation then targets the residual.

- PATH consistently outperforms DR, PLR, and ACCEL across both benchmarks on robustness and generalization, with results averaged over 5 seeds. The evaluation is comprehensive: per-environment robustness breakdowns, generalization on 1000 sampled environments, component ablations disentangling the two phases, and hyperparameter sensitivity analysis.

**Weaknesses**

The prerequisite assumption is strong and untested. The method assumes access to a well-structured curriculum DAG where respecting edge order guarantees efficient progression. This holds by construction in MiniGrid (add blocks) and BipedalWalker (increment parameters), but the paper provides no analysis of what happens when the DAG is misspecified (e.g., when an edge doesn't actually represent a meaningful prerequisite, or when difficulty is non-monotonic along certain paths). In many practical RL domains, the difficulty landscape is complex and multi-dimensional, and a clean DAG may not exist or may be expensive to specify. The paper would benefit from experiments where the DAG is intentionally corrupted or where edges are noisy proxies for true difficulty ordering.

---

> ### Author Rebuttal · Authors · 2026-03-28
>
> We thank Reviewer H6PJ for the thorough review. We address the noted weakness (W) and questions (Q) below.
>
> ---
>
> ### W1 — Strong prerequisite assumption
>
> We respectfully clarify that PATH does not require strict monotonic difficulty along edges. In MiniGrid, $\texttt{NineRoomsFewerDoors}$ and $\texttt{SixteenRoomsFewerDoors}$ (Figure 12) contain fewer blocks yet are harder tasks with lower average solved rates (Table 4). In BipedalWalker, uphill vs. downhill (uphill is generally harder) terrain is not strictly determined by parameters. Despite this, PATH consistently outperforms baselines, remaining robust to non-strictly monotonic DAGs by leveraging underlying curriculum structure.
>
> We further evaluate robustness of PATH itself via **perturbation**, where each successor selection is replaced by a **random** node in the **full** graph space with the probability 10%/30%/50%. We conduct this experiment in both domains
>
> ---
>
> **MiniGrid — Robustness Solved Rate at 10k (Same setting as Table 4)**
>
> |Method|Average Solved Rate|
> |---|---|
> |DR|0.177 ± 0.01|
> |PLR|0.379 ± 0.00|
> |ACCEL|0.557 ± 0.01|
> |PATH (50% perturb)|0.427 ± 0.01|
> |PATH (30% perturb)|0.532 ± 0.01|
> |PATH (10% perturb)|0.674 ± 0.01|
> |**PATH**|**0.762 ± 0.00**|
>
> **BipedalWalker — Mean Robustness Return at 10k (same setting as Table 5)**
>
> |Method|Average Return|
> |---|---|
> |DR|51.60 ± 10.40|
> |PLR|69.47 ± 4.68|
> |ACCEL|142.31 ± 9.71|
> |PATH (50% perturb)|139.06 ± 12.30|
> |PATH (30% perturb)|168.24 ± 8.12|
> |PATH (10% perturb)|184.63 ± 7.28|
> |**PATH**|**198.12** ± **9.69**|
>
> ---
>
> All results are averaged over 5 seeds, and PATH shifts phase at 5k steps. Performance degrades gradually rather than collapsing under perturbation. In MiniGrid, with 30% perturbation, PATH remains comparable to ACCEL; in BipedalWalker, even at 50%, it is comparable with ACCEL and substantially outperforms DR and PLR. Performance degrades gradually rather than catastrophically, demonstrating strong robustness to misspecified DAGs.
>
>
> ---
>
> ### Q1: How PATH handles case when monotonicity is not strictly followed
>
> We thank the reviewer for this question. As discussed above, PATH is robust to noisy/non-monotonic curriculum DAGs in W1. When adding a block creates an easier layout, $\varepsilon_\text{es}$ is met quickly on the successor and the path advances efficiently. Conversely, when encountering a harder successor early, the patience threshold $\varepsilon_\text{p}$ prevents excessive budget allocation.. If harder layout is ultimately solvable, it will be learned when revisited in the future, and PATH’s strategy is to invest budget in more transferable and efficiently learnable progression paths.
>
> ---
> ### Q2 — Scalability in high-dimensional curriculum spaces
>
> Compared to prior automated curriculum methods such as ALP-GMM (Portelas et al., 2019), which require learning a density over the curriculum space, PATH operates via simple random path sampling on the DAG, without any density estimation or model fitting. This leads to substantially lower computational overhead.
>
> The key insight from Sec. 3.4 is that in high-dimensional curriculum spaces, randomly sampled paths exhibit **low overlap**—each path tends to cover a distinct local region (t-ball neighborhood). As a result, PATH naturally maintains strong coverage without suffering from redundancy, making it particularly well-suited for large, high-branching spaces.
>
> Regarding budget allocation, our strategy is empirically guided but consistent across domains. The Random phase is both more efficient (≈2× faster than the active phase, see Appendix B.3) and provides broad early coverage, while the Active phase focuses on exploiting residual unmastered regions to improve the final performance ceiling. Our empirical results support that starting with a random phase until stagnation, followed by a shift to the active phase, is an effective strategy.
>
> ### Limitation — Cost of DAG Specification
>
> PATH is designed for settings where curriculum structure naturally arises, such as parameterized difficulty described in Section 1. In our experiments, DAG construction (Appendix B.1, B.2) is simple and generalizable. We expect that, in similar structured domains (Margolis et al., 2022; Atanassov et al., 2024), the cost of specifying the DAG is trivial compared to its potential benefits for learning across diverse environments.
>
> Moreover, recent work [1] demonstrates that curriculum structures can be automatically generated using LLMs with human priors on difficulties, suggesting that construction can be further relaxed in practice. We agree that developing efficient methods for DAG construction, as well as principled choices of active learning strategies, is an important direction for future work.
>
>
> We will incorporate the above clarifications, limitations, and future directions into the final version of the paper.
>
>
> [1] Liang, William et al. “Eurekaverse: Environment Curriculum Generation via Large Language Models.” Conference on Robot Learning (2024).

---

> > ### Author Rebuttal · Reviewer_H6PJ · 2026-04-03
> >
> > Thank you for the thorough response and the additional perturbation experiments. The graceful degradation under DAG misspecification (rather than catastrophic failure) is reassuring and directly addresses my main concern. I appreciate the authors' willingness to incorporate these clarifications and limitations into the final version.

---

> > > ### Author Response · Authors · 2026-04-06
> > >
> > > Thank you very much for taking the time to read our rebuttal and for indicating that your concern has been fully resolved. We greatly appreciate this. We are also excited to highlight a new benchmark introduced in our response to Reviewer 3WBd’s rebuttal acknowledgement, where we demonstrate that PATH performs well in a non-explicitly parameterized setting, which is more general and better reflects real-world RL problems. We hope the rebuttal and discussion will be helpful as you finalize your overall evaluation and the score of our paper.
> > >
> > > Finally, We thank the reviewer again for the insightful feedback, which has helped strengthen the paper.

---

### Official Review · Reviewer_pN72 · 2026-03-13

**Soundness:** 2
**Presentation:** 3
**Significance:** 3
**Originality:** 2
**Overall Recommendation:** 4
**Confidence:** 2

**Summary:**

This paper proposes PATH, a curriculum learning framework for reinforcement learning that assumes the environment family induces a directed acyclic graph (curriculum DAG) encoding prerequisite relations. Training is framed as active path acquisition over this graph rather than independent environment sampling. PATH operates in two phases: 1) PATH:RANDOM, wich explores diverse curriculum paths to expand coverage. 2) PATH:ACTIVE, which reallocates training budget toward high-regret regions by dispatching additional paths through successor subgraphs. Experiments on MiniGrid and BipedalWalker show improved robustness and generalization compared to domain randomization (DR), Prioritized Level Replay (PLR), and ACCEL.

**Compliance With Llm Reviewing Policy:**

Affirmed.

**Key Questions For Authors:**

Questions:
Q1: How robust is PATH to incorrect DAG structure? If prerequisite edges are noisy or partially wrong, does performance degrade gracefully?

Q2: Can PATH be extended to cyclic or partially ordered curricula? Many real environment families are not strictly DAG-structured.

Q3: What happens when difficulty is non-monotonic? For example, when successor tasks are sometimes easier due to different skill requirements.

**Limitations:**

Limitations not clearly discussed in the paper

**Strengths And Weaknesses:**

Strengths:
1. The key conceptual contribution is reframing curriculum learning from instance selection to path selection. This is meaningful in many RL domains where difficulty structure is explicit.
2. The experimental results show improvements over DR, PLR, and ACCEL on robustness tests, generalization tests, and sampling complexity metrics.

Weaknesses:
1. The method assumes that prerequisite relations are known or easy to construct, difficulty ordering is monotonic along edges, and mastering predecessors reliably reduces cost for successors. These assumptions hold in toy benchmarks but are often non-trivial in real RL problems.
2. PATH’s distinctive element is path-level sampling, but algorithmically it remains close to buffer prioritization plus structured traversal heuristics. The conceptual advance is incremental rather than fundamentally new.

---

> ### Author Rebuttal · Authors · 2026-03-28
>
> We thank Reviewer pN72 for the careful evaluation. We address the noted weaknesses (W) and questions (Q) below..
>
> ---
>
> ### Q1,Q2,Q3 — Robustness to non-strictly monotonic curriculum structure
>
> We thank the reviewer for these questions, which allow us to clarify how PATH performs under imperfect curriculum structures.
>
> **Tolerance to non-monotonicity.**
> We respectfully clarify that PATH does not require strict monotonic difficulty. Specifically, in MiniGrid, $\texttt{NineRoomsFewerDoors}$ and $\texttt{SixteenRoomsFewerDoors}$ (Figure 12) contain fewer placed blocks yet are harder navigation tasks with lower average solved rates (Table 4). This directly demonstrates that PATH effectively handles non-monotonic edges throughout the MiniGrid curriculum. In BipedalWalker, uphill vs. downhill (uphill is generally harder) terrain is not strictly determined by parameters. Despite this, PATH consistently outperforms baselines, remaining robust to non-strictly monotonic DAGs by leveraging underlying curriculum structure.
>
> **Incorrect DAG structure.**
> We further evaluate robustness of PATH via **perturbation**: each successor selection is replaced by a **random** node in the **full** graph space with one probability 10%/30%/50%. We conduct this experiment in both domains
>
> ---
>
> **MiniGrid — Robustness Solved Rate at 10k (Same setting as Table 4)**
>
> |Method|Average Solved Rate|
> |---|---|
> |DR|0.177 ± 0.01|
> |PLR|0.379 ± 0.00|
> |ACCEL|0.557 ± 0.01|
> |PATH (50% perturb)|0.427 ± 0.01|
> |PATH (30% perturb)|0.532 ± 0.01|
> |PATH (10% perturb)|0.674 ± 0.01|
> |**PATH**|**0.762 ± 0.00**|
>
> **BipedalWalker — Mean Robustness Return at 10k (same setting as Table 5)**
>
> |Method|Average Return|
> |---|---|
> |DR|51.60 ± 10.40|
> |PLR|69.47 ± 4.68|
> |ACCEL|142.31 ± 9.71|
> |PATH (50% perturb)|139.06 ± 12.30|
> |PATH (30% perturb)|168.24 ± 8.12|
> |PATH (10% perturb)|184.63 ± 7.28|
> |**PATH**|**198.12** ± **9.69**|
>
> ---
>
> All results are averaged over 5 seeds, and PATH shifts phase at 5k steps. Across both domains, performance degrades gradually rather than collapsing under perturbation. In MiniGrid, with 30% perturbation, PATH remains comparable to ACCEL; in BipedalWalker, even at 50%, it is comparable with ACCEL and substantially outperforms DR and PLR. Performance degrades gradually rather than catastrophically, demonstrating strong robustness to misspecified DAGs.
>
> Furthermore, we note that cyclic curriculum graphs are generally less representative of realistic RL curricula. Cycles can encode bidirectional transferability between tasks. In such cases, PATH is still applicable and we expect it to perform well, as its traversal mechanism does not rely on strict acyclicity.
>
> ---
>
> ### W1, W2 — Nontrivial assumption; contribution is incremental
>
> **Scope of the prerequisite assumption.**
> PATH is designed for settings where curriculum structure naturally arises from environment design, such as parameterized difficulty, compositional edits, or feature-based progression described in Section 1. This is common across many RL domains such as locomotion and navigation (Margolis et al., 2022; Atanassov et al., 2024). Our DAG construction introduces no additional assumptions beyond prior work—it directly leverages the natural complexity increments inherent in each domain (Appendix B.1, B.2).
>
> **Novelty of path**
>
> The key novelty of PATH lies not only in the two-staged learning heuristic, but in introducing a fundamentally different unit of decision for curriculum learning.
>
> Prior works (e.g., ACCEL, PLR, DR) operate at the level of instance selection—deciding which individual environment to sample. In contrast, PATH operates at the level of curriculum traversal—allocating training over paths in a structured environment space.
>
> This distinction is crucial: when environments exhibit inherent structure (e.g., increasing difficulty or compositional variations), treating them as independent instances leads to inefficient exploration and redundant sampling. PATH instead leverages this structure explicitly, enabling training to progress along meaningful trajectories rather than isolated samples.
>
> Such structured difficulty is common in real world RL domains stated in the Remarks of the paper (Margolis et al., 2022; Atanassov et al., 2024), and [1], yet existing methods do not directly model or exploit it. PATH provides a first step toward algorithms that reason over structured curricula rather than flat environment distributions. Moreover, this perspective naturally encourages new research directions, including automatic curriculum graph construction and more principled acquisition strategies for targeting unmastered regions within the curriculum.
>
>
> ---
> We will incorporate the above clarifications, limitations, and discussion of future directions into the final version of the paper.
>
> [1] Liang, William et al. “Eurekaverse: Environment Curriculum Generation via Large Language Models.” Conference on Robot Learning (2024).

---

> > ### Author Rebuttal · Reviewer_pN72 · 2026-04-06
> >
> > Thank you for the thoughtful rebuttal and additional experiments, which help clarify robustness to non-monotonicity and imperfect DAG structure. The perturbation results are encouraging and suggest graceful degradation under noise. Few concerns remain: the reliance on a reasonably well-structured curriculum graph may still limit applicability in more complex or less structured domains, and the methodological novelty remains somewhat incremental relative to prior prioritization-based approaches. I'll keep my score as weak accept.

---

> > > ### Author Response · Authors · 2026-04-06
> > >
> > > We thank reviewer pN72 for the follow-up and positive assessment of our rebuttal. We are glad that the perturbation results help clarify the robustness of PATH to non-monotonicity and imperfect DAG structure.
> > >
> > > We also appreciate your remaining concern regarding applicability in more complex or less structured domains. To further address this point, we conducted an additional experiment in a more general setting using Leaper from Procgen, where difficulty is not explicitly parameterized but instead inferred from observable environment statistics. **The experiment setup and results are described in detail in our response to the rebuttal acknowledgement by Reviewer 3WBd, and we respectfully refer the reviewer to that response for an additional example of how PATH can be implemented in one general and applicable setting.** The DAGs we construct for MiniGrid, BipedalWalker, and Leaper are all imperfect, yet our algorithm still achieves meaningful results. Since PATH is one lightweight and efficient automated curriculum learning algorithm, we believe that the value of PATH extends beyond highly structured settings: even under limited assumptions and in less structured environment sets, it remains worthwhile to explore its ability to exploit implicit curriculum structure with beneficial training potential.
> > >
> > > Thank you again for your detailed feedback, which has greatly helped improve our paper.

---

### Official Review · Reviewer_3WBd · 2026-03-13

**Soundness:** 3
**Presentation:** 3
**Significance:** 3
**Originality:** 3
**Overall Recommendation:** 5
**Confidence:** 3

**Summary:**

This paper formulates curriculum learning in RL as active learning over an explicit curriculum DAG, where training acquires informative paths rather than isolated environments, and proposes PATH, a two-stage method with random path exploration followed by regret-driven active refinement over successor regions. The paper instantiates such DAGs from edit-based and parameter-based environment families, and reports stronger robustness and held-out generalization than DR, PLR, and ACCEL on MiniGrid and BipedalWalker. Ablations support the random-then-active design.

**Compliance With Llm Reviewing Policy:**

Affirmed.

**Final Justification:**

The authors have addressed my concerns during the rebuttal. I appreciate the added evaluation environment.

**Key Questions For Authors:**

1. How robust is PATH to noisy or misspecified curriculum DAGs, or to cases where prerequisite relations are only partially valid?
2. Since MiniGrid never switches to PATH:ACTIVE by 20k updates, can you provide longer-horizon or alternative-switch evidence showing the active phase matters there too? That would strengthen the claim that the full PATH method, not only PATH:RANDOM, works across domains.

**Limitations:**

Partially yes. In the conclusion, the paper could more explicitly discuss failure modes when the DAG is expensive to construct, misspecified, or does not faithfully encode prerequisite structure.

**Strengths And Weaknesses:**

Strengths:
* The formulation is clear and practically motivated: making ACCEL-style local edits explicit as a curriculum DAG and learning over paths is a useful perspective, and Figure 1/Algorithm 1 communicate the method well. This is a nice combination of curriculum structure and replay-based prioritization.
* The empirical gains are substantial, especially on BipedalWalker: Results show a large robustness improvement over ACCEL and better held-out return/solve rate. An ablation supports that PATH is better than PATH:Random or PATH:Active alone.

Weaknesses:
* In my view, the main soundness gap is that the central analysis in Section 3.4 is heuristic rather than rigorous. Roughly uniform mastery cost along paths, regret indicating useful successor regions, and local "coverage" via graph neighborhoods are plausible assumptions but not tested empirically, so the paper shows efficacy more than it explains when PATH should work.
* The authors claim to assess a broad domain, but the evidence is still limited to two curated benchmarks with hand-constructed curriculum DAGs. On MiniGrid the paper states PATH never leaves PATH:RANDOM within 20k updates, so the full two-stage story is only really validated on one benchmark; and absolute BipedalWalker generalization remains somewhat modest. The paper could for example additionally consider a 3D embodied continuous-control task with a natural curriculum graph such as robotics-style navigation or locomotion.

---

> ### Author Rebuttal · Authors · 2026-03-28
>
> We thank Reviewer 3WBd for the comprehensive assessment. We address the noted weakness (W) and questions (Q) below..
>
> ---
>
> ### W1 — Section 3.4 is heuristic and lacks empirical validation
>
> We respectfully clarify that Section 3.4 is directly supported by ablations in Section 4.3.
>
> (1) PATH:RANDOM achieves large coverage early, then saturates.
> This is explicitly observed in Fig. 10b: PitGap sampling complexity increases rapidly during the first 8k updates, and then plateaus with increasing repetition.
>
> (2) PATH:ACTIVE targets the residual unmastered region after saturation.
> In Fig. 10b, PATH:ACTIVE continues increaseing sampling complexity steadily throughout 20k updates, including well beyond the PATH:RANDOM plateau.
>
> (3) Coverage illustration in Figure 2.
> Fig. 11 directly validates this: PATH achieves a higher generalization solved rate.
>
> ---
>
> ### W2 and Q2 — Limited benchmarks, lack of two phases algorithm in MiniGrid, and modest performance on Bipedalwalker generalization
>
> - On benchmarks. We follow established prior work ACCEL, ADD by Chung et al., and evaluate on MiniGrid and BipedalWalker, which span complementary RL regimes—sparse vs. dense reward, discrete vs. continuous control, and edit-based vs. parameter-based curriculum structures. The consistent gains across both domains suggest that PATH captures a general curriculum principle rather than a domain-specific heuristic. We agree that extending to more realistic environments is an important direction for future work.
>
> - **On the necessity of the second phase in MiniGrid**. While PATH:RANDOM alone achieves strong performance, we explicitly test the contribution of the active phase by shortening the horizon (5 seeds, 10k updates, with phase switch at 5k):
>
> |Method|Average Solved Rate (5k)|Average Solved Rate (10k)|
> |---|---|---|
> |DR|0.166 ± 0.01|0.177 ± 0.01|
> |PLR|0.187 ± 0.01|0.379 ± 0.00|
> |ACCEL|0.303 ± 0.01|0.557 ± 0.01|
> |PATH:ACTIVE|0.326 ± 0.01|0.588 ± 0.01|
> |PATH:RANDOM|0.345 ± 0.00|0.697 ± 0.01|
> |**PATH**|**0.345 ± 0.00**|**0.761 ± 0.00**|
>
> This test has the same setting as Table 4. Comparing results from 5k to 10k updates, PATH (0.761) exceeds both PATH:RANDOM (0.697) and PATH:ACTIVE (0.588), confirming that the active phase provides additional gains.
>
> - On BipedalWalker generalization. We acknowledge that improvements in the full parameter space appear more modest, largely because a significant portion of the environment space is inherently unsolvable. To better isolate meaningful differences, we restrict the parameter space from \([10,10,5,5,9]\) to a more tractable range \([6,6,3,3,6]\), while still evaluating on 1000 sampled environments.
>
> |Method|Mean Return|Solved Rate|
> |---|---|---|
> |DR|42.22 ± 10.58|0.04 ± 0.02|
> |PLR|66.65 ± 8.22|0.10 ± 0.02|
> |ACCEL|121.16 ± 9.69|0.28 ± 0.03|
> |PATH|**201.14 ± 6.45**|**0.55 ± 0.02**|
>
> We still observe a consistent trend: PATH achieves nearly 2× improvement over ACCEL in both mean reward and solved rate at 20k updates (averaged over 5 seeds).
>
> ---
>
> ### Q1 — Whether Path has robustness to noisy or misspecified curriculum DAGs
>
> We respectfully clarify that PATH does not assume strict monotonicity.
>
> Empirically, both domains exhibit non-strict monotonicity (e.g., in MiniGrid, some layouts with fewer blocks can still be harder (Figure 12, Table 4); in BipedalWalker, uphill vs. downhill terrain is not strictly determined by parameters). Despite this, PATH consistently outperforms baselines, remaining robust to non-strictly monotonic DAGs by leveraging underlying curriculum structure.
>
> We further evaluate robustness of PATH via **perturbation**: each successor selection is replaced by a **random** node in the **full** graph space with one probability 10%/30%/50%. We conduct this experiment in both domains
>
> **MiniGrid — Robustness Solved Rate at 10k (Same setting as Table 4)**
>
> |Method|Average Solved Rate|
> |---|---|
> |DR|0.177 ± 0.01|
> |PLR|0.379 ± 0.00|
> |ACCEL|0.557 ± 0.01|
> |PATH (50% perturb)|0.427 ± 0.01|
> |PATH (30% perturb)|0.532 ± 0.01|
> |PATH (10% perturb)|0.674 ± 0.01|
> |**PATH**|**0.762 ± 0.00**|
>
> **BipedalWalker — Mean Robustness Return at 10k (same setting as Table 5)**
>
> |Method|Average Return|
> |---|---|
> |DR|51.60 ± 10.40|
> |PLR|69.47 ± 4.68|
> |ACCEL|142.31 ± 9.71|
> |PATH (50% perturb)|139.06 ± 12.30|
> |PATH (30% perturb)|168.24 ± 8.12|
> |PATH (10% perturb)|184.63 ± 7.28|
> |**PATH**|**198.12** ± **9.69**|
>
> ---
>
> All results are averaged over 5 seeds, and PATH shifts phase at 5k steps. Across both domains, performance degrades gradually rather than collapsing under perturbation. In MiniGrid, with 30% perturbation, PATH remains comparable to ACCEL; in BipedalWalker, even at 50%, it is comparable with ACCEL and substantially outperforms DR and PLR. Performance degrades gradually rather than catastrophically, demonstrating strong robustness to misspecified DAGs.
>
> ---
> We will incorporate the above clarifications, limitations, and future directions into the final version of the paper.

---

> > ### Author Rebuttal · Reviewer_3WBd · 2026-04-04
> >
> > Thank you for your rebuttal, based on which I will increase my score to recommend acceptance.
> > However, as indicated in my original review, I do still believe that the empirical support could be strengthened by evaluating on a third, more complex environment.

---

> > > ### Author Response · Authors · 2026-04-06
> > >
> > > We sincerely appreciate Reviewer 3WBd’s willingness to increase the score. We are glad that our rebuttal was able to address most of your concerns.
> > >
> > > Relevant to your concern about including a third benchmark, we conducted a new experiment on Leaper from Procgen[1]. This benchmark directly addresses the setting without explicit parameterization given a batch of diverse RL environments: the environments do not provide an obvious editing mechanism or controllable parameters, making them more representative of realistic scenarios.
> > >
> > > ## Leaper as a non-explicit parameterization case.
> > >
> > > Leaper is a Procgen game in which a frog agent must cross a road (dodging cars) and a river (hopping on logs) to reach a goal lane. The state space is represented by 64×64x3 pixel observations, and the action space follows 15-dimensional Procgen game setting.
> > > It is a sparse-reward problem: the agent receives a reward of either 0 or 10 upon success. Unlike MiniGrid or BipedalWalker, Leaper does not expose an explicit editing structure or difficulty parameter. Instead, we define difficulty as emerging naturally from four observable environmental axes: (1) maximum car speed, (2) maximum log speed, (3) number of road lanes, and (4) number of water lanes.
> > >
> > > ## Applying PATH.
> > >
> > > We assume that increasing each axis leads to a harder environment, which provides a natural difficulty ordering. Based on this prior, we treat each axis as a curriculum dimension and assume monotonic difficulty along it. This is how PATH is applied in the implicit case: by identifying observable structure and using it as a reasonable prior, without requiring a strictly guaranteed monotone DAG.
> > >
> > > We represent the curriculum DAG using difficulty buckets, where each bucket is a 4-dimensional tuple. We randomly sample 300 Leaper environments as the fixed training set. To conveniently construct the buckets, we leverage CProcgen [1], a C++ Procgen backend that exposes the context parameters of each Procgen environment, allowing us to directly read out the four difficulty axes. We emphasize that this step can, in principle, be performed without explicit parameterization—observable environment statistics are sufficient. We compute statistics for each dimension and then discretize each axis into 3 levels (easy/medium/hard), with an approximately equal number of environments per level, yielding 3^4 = 81 buckets. For instance, the number of car lanes ranges from 1 to 3 across the three difficulty levels along that dimension. The induced DAG starts at bucket [0,0,0,0]; once PATH’s early-stop criterion is satisfied, it randomly selects a neighboring bucket with one dimension incremented and samples uniformly from the environments within that bucket.
> > >
> > > ## Experiment Configuration.
> > >
> > > We use 300 training environments across 81 buckets, 32 parallel workers for PPO, 1000 total PPO update steps, rollout length 256, replay buffer size 30, early-stop threshold 10.0, patience of 20 updates, and a phase shift after 500 PPO updates for PATH. We adopt the PPO hyperparameter settings from the PLR paper. We compare PATH, ACCEL, PLR, and DR across 5 seeds. Each run requires approximately one hour of training.
> > >
> > > ## Results
> > >
> > > | Method | Solved Rate |
> > > |--------|-------------|
> > > | DR | 0.568 ± 0.130 |
> > > | PLR | 0.446 ± 0.082 |
> > > | ACCEL | 0.512 ± 0.162 |
> > > | PATH | 0.655 ± 0.042 |
> > >
> > > (Evaluated on 100 held-out test environments and averaged over 5 seeds.)
> > >
> > > PATH achieves a clear advantage over all baselines, with a higher average solved rate and a significantly lower standard error. One interesting observation is that DR is a very strong baseline. We believe this is due to the relatively small training set, where uniform sampling can already provide strong coverage. However, PATH still outperforms DR. This demonstrates that even when the curriculum DAG is not explicitly given, but instead constructed from observable difficulty axes, PATH’s structured traversal can still yield meaningful gains by leveraging the transferability of implicit curricula. Full experimental details, including training curves, will be incorporated into the final version of the paper.
> > >
> > > We also acknowledge one limitation of PATH is that existing evaluations are mostly conducted in common benchmarks, and extending PATH to more realistic and complex settings is an important future direction. In particular, we see strong potential in combining PATH with goal-conditioned tasks for robotic arm manipulation, where both goal variables and environment parameters can serve as natural difficulty axes.
> > >
> > > Finally, we would like to sincerely thank you again for your constructive feedback, which has helped improve our paper.
> > >
> > > ---
> > >
> > > [1] Cobbe, K., Hesse, C., Hilton, J., and Schulman, J. Leveraging Procedural Generation to Benchmark Reinforcement Learning. ICML 2020.
> > >
> > > [2] Tan, Z., Wang, K., Wang, X., 2023. C-Procgen: Empowering Procgen with Controllable Contexts.

---

### Decision · Program_Chairs · 2026-04-30

**Decision:**

Accept (regular)

**Comment:**

This paper frames curriculum learning as active path acquisition on a graph to better leverage environment structure. The central takeaway is that treating curricula as paths significantly improves robustness and generalization across sparse and dense reward settings. Reviewers mostly liked the simple yet effective conceptual shift and the strong empirical results on MiniGrid and BipedalWalker. One reviewer stayed concerned about baseline variety and DAG dependencies, but the authors provided a new Procgen benchmark and noise robustness tests that most found convincing. The rebuttal clarified notation issues and showed the method works with different properties for task difficulty. I recommend acceptance.